# Fibroblast-expressed LRRC15 is a receptor for SARS-CoV-2 spike and controls antiviral and antifibrotic transcriptional programs

**Lipin Loo**[1ⓞ], **Matthew A. Waller**[1ⓞ], **Cesar L. Moreno**[1], **Alexander J. Cole**[2], **Alberto Ospina Stella**[3], **Oltin-Tiberiu Pop**[4], **Ann-Kristin Jochum**[4,5], **Omar Hasan Ali**[4,6,7], **Christopher E. Denes**[1], **Zina Hamoudi**[1], **Felicity Chung**[1], **Anupriya Aggarwal**[3], **Jason K. K. Low**[8], **Karishma Patel**[8], **Rezwan Siddiquee**[8], **Taeyoung Kang**[9], **Suresh Mathivanan**[9], **Joel P. Mackay**[8], **Wolfram Jochum**[5], **Lukas Flatz**[4,10], **Daniel Hesselson**[2], **Stuart Turville**[3], **G. Gregory Neely**[1]*

1 Charles Perkins Centre, Dr. John and Anne Chong Lab for Functional Genomics, Centenary Institute, and School of Life and Environmental Sciences, University of Sydney, Camperdown, New South Wales, Australia, 2 Centenary Institute and Faculty of Medicine and Health, The University of Sydney, Sydney, New South Wales, Australia, 3 The Kirby Institute, University of New South Wales, New South Wales, Australia, 4 Institute for Immunobiology, Kantonsspital St. Gallen, St. Gallen, Switzerland, 5 Institute for Pathology, Kantonsspital St. Gallen, St. Gallen, Switzerland, 6 Department of Medical Genetics, Life Sciences Institute, University of British Columbia, Vancouver, British Columbia, Canada, 7 Department of Dermatology, University Hospital Zurich, University of Zurich, Zurich, Switzerland, 8 School of Life and Environmental Sciences, The University of Sydney, Sydney, New South Wales, Australia, 9 Department of Biochemistry and Genetics, La Trobe Institute for Molecular Science, La Trobe University, Melbourne, Victoria, Australia, 10 Center for Dermatooncology, Department of Dermatology, Eberhard Karls University of Tübingen, Tübingen, Germany

ⓞ These authors contributed equally to this work.
* greg.neely@sydney.edu.au

**Data Availability Statement:** CRISPR screen raw read counts have been deposited at GSE186475 and are publicly available (https://www.ncbi.nlm.

## Abstract

Although ACE2 is the primary receptor for Severe Acute Respiratory Syndrome Coronavirus 2 (SARS-CoV-2) infection, a systematic assessment of host factors that regulate binding to SARS-CoV-2 spike protein has not been described. Here, we use whole-genome CRISPR activation to identify host factors controlling cellular interactions with SARS-CoV-2. Our top hit was a *TLR*-related cell surface receptor called *leucine-rich repeat-containing protein 15* (*LRRC15*). *LRRC15* expression was sufficient to promote SARS-CoV-2 spike binding where they form a cell surface complex. *LRRC15* mRNA is expressed in human collagen-producing lung myofibroblasts and LRRC15 protein is induced in severe Coronavirus Disease 2019 (COVID-19) infection where it can be found lining the airways. Mechanistically, LRRC15 does not itself support SARS-CoV-2 infection, but fibroblasts expressing LRRC15 can suppress both pseudotyped and authentic SARS-CoV-2 infection in *trans*. Moreover, LRRC15 expression in fibroblasts suppresses collagen production and promotes expression of IFIT, OAS, and MX-family antiviral factors. Overall, LRRC15 is a novel SARS-CoV-2 spike-binding receptor that can help control viral load and regulate antiviral and antifibrotic transcriptional programs in the context of COVID-19 infection.

nih.gov/geo/query/acc.cgi?acc=GSE186475).
CRISPR screen analysis is shown in Figs 2 and S2.
CRISPR screen output is reported in S1 Table. RNA
sequencing bam files have been deposited on NCBI
SRA at PRJNA895078 and are publicly available
(https://www.ncbi.nlm.nih.gov/bioproject/
PRJNA895078). Analysis of RNA sequencing data
is shown in Fig 6. Results of differential gene
expression analysis are reported in S5 Table.
Canonical Pathways output from Ingenuity
Pathway Analysis is reported in S6 Table. This
paper also analyses existing publicly available
single cell RNA-sequencing data (GSE158127,
SCP1052, SCP1219). All data reported in this paper
will be shared by the lead contact upon request.
This paper does not report original code. Any
additional information required to reanalyse the
data reported in this paper is available from the
lead contact upon request. All underlying data in
figures can be found at DOI: 10.5281/zenodo.
7416876.

**Funding:** This work was supported by the National
Health and Medical Research Council
(APP1107514, APP1158164, APP1158165 to G.G.
N), University of Sydney (Drug Discovery Initiatve
Seed Funding and Dr John and Anne Chong
Fellowship to L.L.), Schweizerischer Nationalfonds
zur Förderung der Wissenschaftlichen Forschung
(PP00P3_157448 to L.F., P400PM_194473 to O.H.
A.). The funders had no role in study design, data
collection and analysis, decision to publish, or
preparation of the manuscript.

**Competing interests:** The authors declare no
competing interests.

**Abbreviations:** ACE2, angiotensin-converting
enzyme 2; COVID-19, Coronavirus Disease 2019;
CRISPRa, CRISPR activation; DEG, differentially
expressed gene; FFPE, formalin-fixed paraffin
embedded; gDNA, genomic DNA; HE, hematoxylin
and eosin; IFIT, interferon-induced proteins with
tetratricopeptide repeats; IMAC, immobilized metal
affinity chromatography; LRRC15, leucine-rich
repeat-containing protein 15; MALLS, multiple
angle laser light scattering; MFI, mean fluorescence
intensity; MOI, multiplicity of infection; NTC, non-
targeting control; PAMP, pathogen-associated
molecular pattern; SARS-CoV-2, Severe Acute
Respiratory Syndrome Coronavirus 2; sgRNA,
single-guide RNA; TLR, toll-like receptor;
TMPRRS2, Transmembrane serine protease 2;
WT, wild-type.

## Introduction

The Coronavirus Disease 2019 (COVID-19) pandemic, caused by Severe Acute Respiratory Syndrome Coronavirus 2 (SARS-CoV-2), represents the greatest public health challenge of our time. As of October 2022, there have been over 620,000,000 reported cases of COVID-19 globally and more than 6,500,000 deaths (WHO). SARS-CoV-2 shows high sequence similarity (79.6%) with Severe Acute Respiratory Syndrome Coronavirus (SARS-CoV-1), and because of this similarity, angiotensin-converting enzyme 2 (ACE2), the primary entry receptor for SARS-CoV-1, was quickly identified as the SARS-CoV-2 spike receptor [1–4]. However, a comprehensive search for other host factors that promote SARS-CoV-2 spike binding has not yet been reported.

To identify novel host factors that can influence cellular interactions with the SARS-CoV-2 spike protein, we used a whole-genome CRISPR activation approach. Using the Calabrese Human CRISPR Activation Pooled Library [5], we identified a TLR-related cell surface receptor named leucine-rich repeat-containing protein 15 (LRRC15) as a novel SARS-CoV-2 spike-binding protein in 3 independent whole-genome screens and confirmed this interaction via flow cytometry, immunoprecipitation, and confocal microscopy. *LRRC15* is primarily expressed in innate immune barriers including placenta, skin, and lymphatic tissues as well as perturbed-state tissue fibroblasts, and we found LRRC15 protein is absent in control lungs, but highly expressed in COVID-19 patients, where it lines the airways. Mechanistically, LRRC15 is not a SARS-CoV-2 entry receptor but can antagonize SARS-CoV-2 infection of ACE2$^+$ cells when expressed on nearby cells. At the cellular level, LRRC15 is expressed in fibroblasts and these cells increase with COVID-19 infection. Moreover, by RNAseq, we found expression of LRRC15 drives a specific antiviral response in fibroblasts while suppressing collagen gene expression. In summary, we show LRRC15 physically links SARS-CoV-2 to perturbed-state fibroblasts, where LRRC15 expression can control the balance between fibrosis and antiviral responses, and this activity may help promote COVID-19 resolution while preventing COVID-19 lung fibrosis.

## Results

### High-throughput SARS-CoV-2 spike-binding assay

Based on a priori knowledge of SARS-CoV-1, ACE2 was rapidly identified as the primary receptor for SARS-CoV-2 spike protein [3]. To investigate other host factors that modulate cellular interactions with SARS-CoV-2 spike, we employed a pooled CRISPR activation (CRISPRa) screening approach. To this end, we developed a novel cellular flow cytometry-based SARS-CoV-2 spike-binding assay using Alexa Fluor 488-labeled spike protein (Spike488; **Fig 1A**). While wild-type HEK293T (WT HEK293T) cells that express low levels of ACE2 show minimal binding to Spike488, when we provided *ACE2* cDNA HEK293T-*ACE2* cells exhibited high Spike488-binding activity (**Fig 1B**). To assess the sensitivity of this assay, we mixed HEK293T-*ACE2* and WT HEK293T cells at various ratios and then measured Spike488 binding by flow cytometry. An increase in Spike488-binding cells could be detected when as little as 1% of the total population was ACE2$^+$, indicating that this assay has sufficient sensitivity to enable genome-wide screens (**Fig 1C**). To perform a pooled CRISPRa screen with this system, we generated a stable HEK293T cell line expressing CRISPRa machinery (*MS2-p65-HSF + VP64*; HEK293T-CRISPRa) (**Fig 1D**). We tested HEK293T-CRISPRa clones for the ability to induce *ACE2* expression using 3 independent single-guide RNAs (sgRNAs) [6]. We selected Clone 1 for further use, since it induced similar levels of *ACE2* expression compared to cDNA overexpression, (**S1A Fig**), and confirmed that CRISPRa induction of ACE2 expression conferred Spike488 binding by flow cytometry (**Fig 1E**).

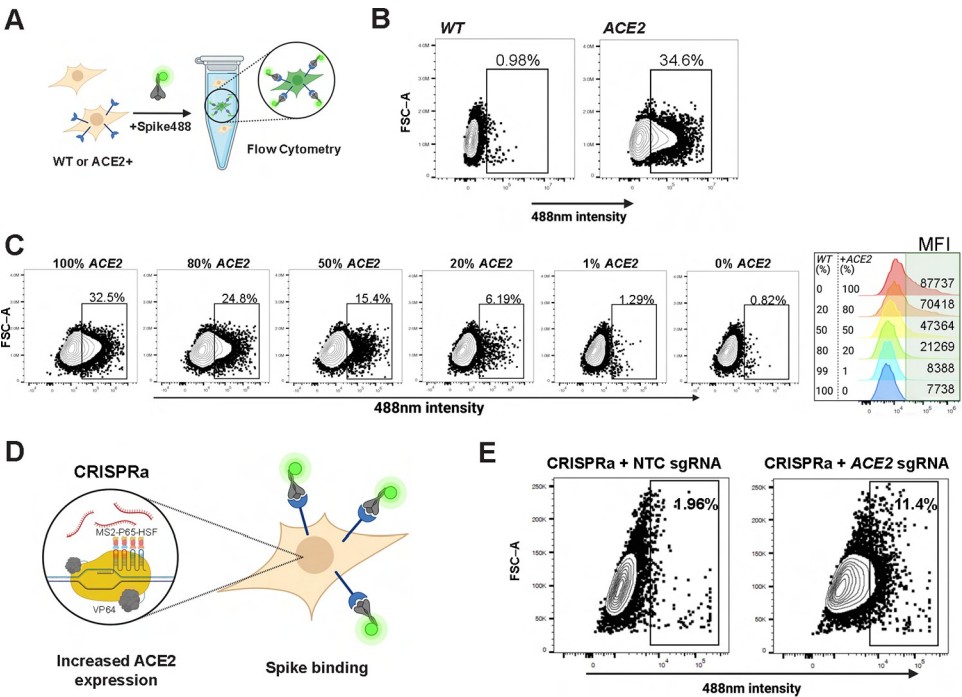

**Fig 1. A sensitive FACS-based SARS-CoV-2 spike-binding assay amenable to high-throughput screening.** (**A**) Schematic of SARS-CoV-2 spike-binding assay. HEK293T cells with stable integration of *ACE2* cDNA for overexpression (HEK293T-*ACE2*) are incubated with Alexa Fluor 488-conjugated SARS-CoV-2 spike protein (Spike488). Spike488-binding cells are then detected by flow cytometry. (**B**) Representative flow cytometry plots for *WT* HEK293T and HEK293T-*ACE2* incubated with Spike488 ($N = 3$). See also **S1B Fig** for gating strategy. (**C**) Titration of HEK293T-*ACE2* (*ACE2*) cells with *WT* HEK293T cells. Approximately 1% HEK293T-*ACE2* cells showed a difference to baseline non-specific binding. Histogram summary showing MFI of flowed cells. (**D**) Schematic of CRISPRa system used. (**E**) Representative plot of flow cytometry for a clonal HEK293T-CRISPRa cell line transduced with NTC sgRNA or *ACE2* sgRNA (expression confirmation via RT-qPCR in **S1A Fig**). The data underlying all panels in this figure can be found in DOI: 10.5281/zenodo.7416876. ACE2, angiotensin-converting enzyme 2; CRISPRa, CRISPR activation; MFI, mean fluorescence intensity; NTC, non-targeting control; SARS-CoV-2, Severe Acute Respiratory Syndrome Coronavirus 2; sgRNA, single-guide RNA.

## CRISPR activation screening for regulators of SARS-CoV-2 spike binding identifies LRRC15

Having established the utility of our system, we used the Calabrese Human CRISPR Activation Pooled guide Library [5] to drive CRISPRa-dependent expression of the human genome in HEK293T-CRISPRa cells. Cells were infected with lentivirus-packaged CRISPRa sgRNAs and then selected on puromycin to enrich for transduced cells. Transduced cells were incubated with Spike488 and sorted by FACS to isolate CRISPRa-*sgRNA* cells with enhanced spike binding. Overall, pooled CRISPRa-*sgRNA* cells showed more spike binding than mock-transduced controls (**S1B and S1C Fig**). Genomic DNA (gDNA) was collected from unselected or Spike488-selected cells and sgRNA abundance quantified by sequencing (**Fig 2A**) and then data analyzed using the MAGeCK analysis platform (v0.5.9.2) [7] and plotted using MAGeCKFlute (v1.12.0) [8]. Our top hit was the transmembrane protein LRRC15 (LogFC 4.748, *P* value $2.62 \times 10^{-7}$, FDR 0.00495), followed by the SARS-CoV-2 entry receptor ACE2 (LogFC 2.1343, *P* value $2.65 \times 10^{-5}$, FDR 0.25). (**Fig 2B–2D** and **S1 Table**). Moreover, we conducted 2 additional screens under slightly different conditions, and in all screens our top hit was LRRC15 (**S2A–S2F Fig**).

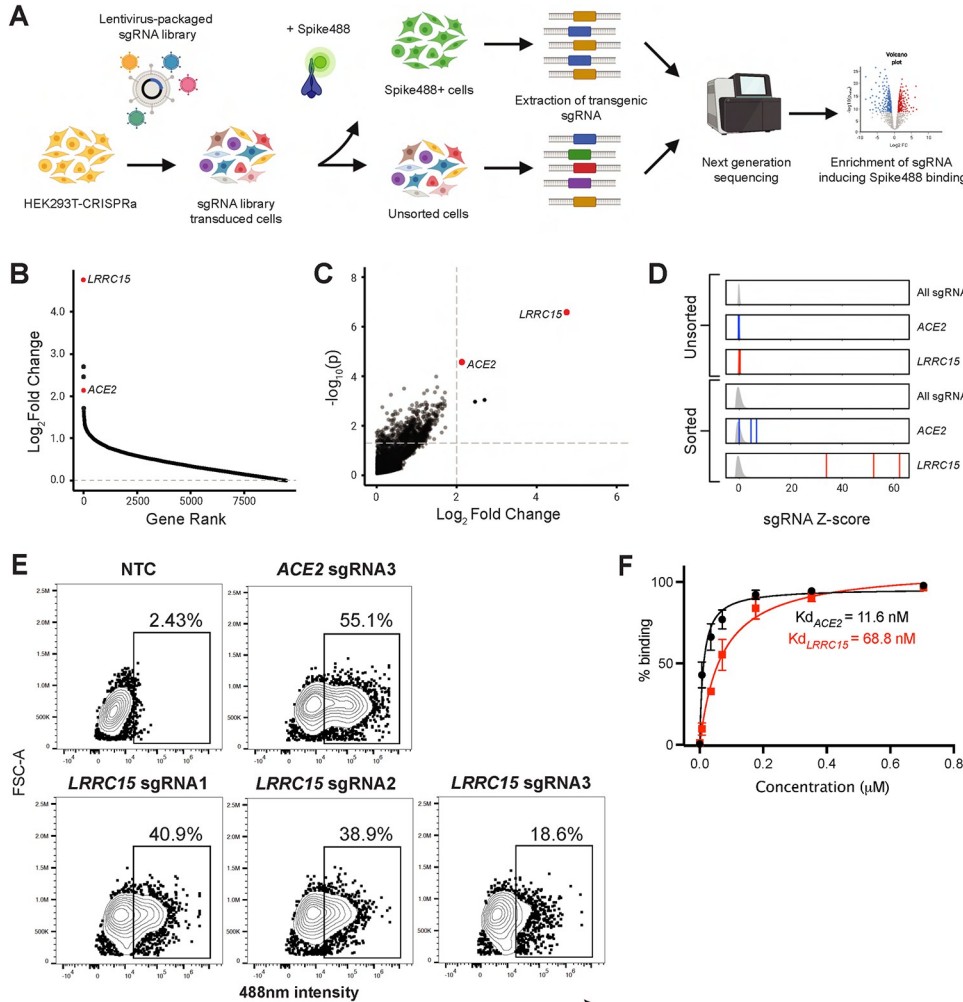

**Fig 2. Whole-genome CRISPRa screening identified LRRC15 as a novel SARS-CoV-2 spike-binding protein. (A)**
Schematic of CRISPRa screen used to identify regulators of SARS-CoV-2 spike binding. **(B)** Ranking of all genes in
screen 1 by log$_2$ fold change (LFC) calculated using MAGeCK and plotted using MAGeCKFlute. See also **S1 Table**. **(C)**
Gene enrichment analysis of screen 1 performed using MAGeCK. Horizontal dotted line indicates *p*-value = 0.05.
Vertical dotted line indicates LFCs of 2. *P*-values and LFCs for all genes are reported in **S1 Table**. **(D)** sgRNA Z-scores
for screen 1 unsorted and sorted samples. Density curve for all sgRNA Z-scores in sample (i.e., sorted or unsorted) is
shown in gray. Z-scores for each guide are indicated by vertical lines (blue ACE2, red LRRC15). **(E)** Flow cytometry
analysis of HEK293T-CRISPRa cells transduced with 3 independent *LRRC15* sgRNAs. HEK293T-CRISPRa transduced
with *ACE2* sgRNA3 were used as a positive control and NTC sgRNA-transduced HEK293T-CRISPRa cells were used
as a negative control (*N* = 3). **(F)** Quantification of Spike647 binding in *ACE2* sgRNA3 and *LRRC15* sgRNA1 cells via
flow cytometry. Dissociation constant (Kd) was calculated by fitting with nonlinear regression (1 site—specific
binding). *N* = 3, error bars represent SD. The data underlying all panels in this figure can be found in DOI: 10.5281/
zenodo.7416876. ACE2, angiotensin-converting enzyme 2; CRISPRa, CRISPR activation; LFC, log2 fold change;
LRRC15, leucine-rich repeat-containing protein 15; NTC, non-targeting control; SARS-CoV-2, Severe Acute
Respiratory Syndrome Coronavirus 2; sgRNA, single-guide RNA.

We expressed the *LRRC15* sgRNAs that were hits in our screens in HEK293T-CRISPRa
cells and confirmed that they induce expression of *LRRC15* (approximately 2,000-fold induc-
tion, **S2G Fig**). Moreover, LRRC15-overexpressing cells dramatically increased SARS-CoV-2
Spike488 binding, with *LRRC15* sgRNA 1 inducing binding to levels comparable to cells over-
expressing *ACE2* sgRNA3 (**Fig 2E**). LRRC15 overexpression did not itself up-regulate *ACE2*
transcription, suggesting the increased spike binding in LRRC15-expressing cells is

independent of ACE2 up-regulation (**S2H Fig**). Conversely, only 1 of the 3 *ACE2* sgRNAs from the Calabrese library efficiently activated *ACE2* expression (**S2I and S2J Fig**), explaining why *ACE2* itself was not a higher ranked hit in our 3 CRISPRa screens (**Figs 2D and S2A– S2D**). To avoid spectral overlap with GFP-expressing cell lines, we conjugated spike with Alexa Fluor 647 (Spike647), which was used for the rest of the study. Using *ACE2* sgRNA3 and *LRRC15* sgRNA1 cells, we measured 11.6 nM affinity for ACE2/Spike647, which is similar to previous estimates (range: 4.7 to 133.3 nM [9–11]) and 68.8 nM for LRRC15/Spike647 (**Fig 2F**).

## LRRC15 is a new transmembrane SARS-CoV-2 spike receptor

LRRC15 is a 581 amino acid (a.a.) leucine-rich repeat (LRR) protein with 15 extracellular LRRs followed by a single transmembrane domain and a short 22 a.a. intracellular domain (**Fig 3A and 3B**). LRRC15 belongs to the LRR Tollkin subfamily that includes TLR1-13 and is most closely related to the platelet von Willebrand factor receptor subunit Glycoprotein V (GP5) [12] (**Fig 3C**, full tree in **S3A Fig**). To confirm a role for LRRC15 in SARS-CoV-2 spike binding and ensure the interaction was not an artifact of our CRISPRa strategy, we transfected *LRRC15-GFP* cDNA into HEK293T cells and observed Spike647 binding by flow cytometry. There are 2 reported isoforms of LRRC15 (LRRC15_1 and LRRC15_2), with LRRC15_1 having 6 additional amino acids at the N-terminus. Although cells transfected with *GFP* alone showed no binding to Spike647, cells expressing *LRRC15* isoform 1 or 2 both showed strong spike binding (**Fig 3D**). While LRRC15-dependent spike binding was higher than cells stably expressing ACE2 (62.1% and 64.5% versus 48.8%), co-expression of LRRC15 with ACE2 was additive resulting in 86.3% positive (LRRC15_1) or 83.8% positive (LRRC15_2) cells (**Fig 3E**). Cells stably expressing known SARS-CoV-2 receptor ACE2 and the spike-priming protease TMPRSS2 bound Spike647 regardless of LRRC15 expression (**Fig 3F**). However, LRRC15 expression in HEK293T-*ACE2-TMPRSS2* cells still enhanced the amount of cell surface Spike647 bound by each cell as measured by mean fluorescence intensity (MFI) (**Fig 3G**). Moreover, both *LRRC15* isoforms colocalized with Spike647 (**Fig 3H**). To independently confirm an interaction between LRRC15 and SARS-CoV-2 spike protein, we added spike to LRRC15-expressing cells, immunoprecipitated LRRC15, then blotted for both LRRC15 and spike. While control *GFP*-transfected HEK293T cells did not show any signal at the size predicted for Spike (approximately 200 kDa [13]) (**S3B and S3C Fig**), when we pulled down either LRRC15_1 or LRRC15_2, in both cases we co-immunoprecipitated spike protein in the eluate (**Fig 3I**). Taken together, these data show that LRRC15 expression is sufficient to confer SARS-CoV-2 spike binding to HEK293T cells, and LRRC15 can further enhance spike interactions in the presence of ACE2 and TMPRSS2.

## LRRC15 is not a SARS-CoV-2 entry receptor but can suppress spike-mediated entry

We next asked if LRRC15 can act as a receptor for SARS-CoV-2 and mediate viral entry. For this, we used a SARS-CoV-2 pseudotyped lentivirus system (SARS-CoV-2 pseudovirus) that displays the SARS-CoV-2 spike protein and carries a luciferase reporter. LRRC15 did not confer SARS-CoV-2 pseudovirus tropism in WT HEK293T (**S4A Fig**) and HEK293T-*ACE2* cells across a wide range of LRRC15 or pseudovirus doses (**S4B Fig**). We then tested if LRRC15 expression impacted infection of HEK293T-*ACE2* and HEK293T-*ACE2-TMPRSS2* cells. Indeed, LRRC15 expression in HEK293T-*ACE2* or HEK293T-*ACE2-TMPRSS2* cells show a relatively strong ability to suppress SARS-CoV-2 pseudovirus infection (**S4C and S4D Fig**). Next, we tested if LRRC15 expression can also suppress viral replication and cytopathic effect

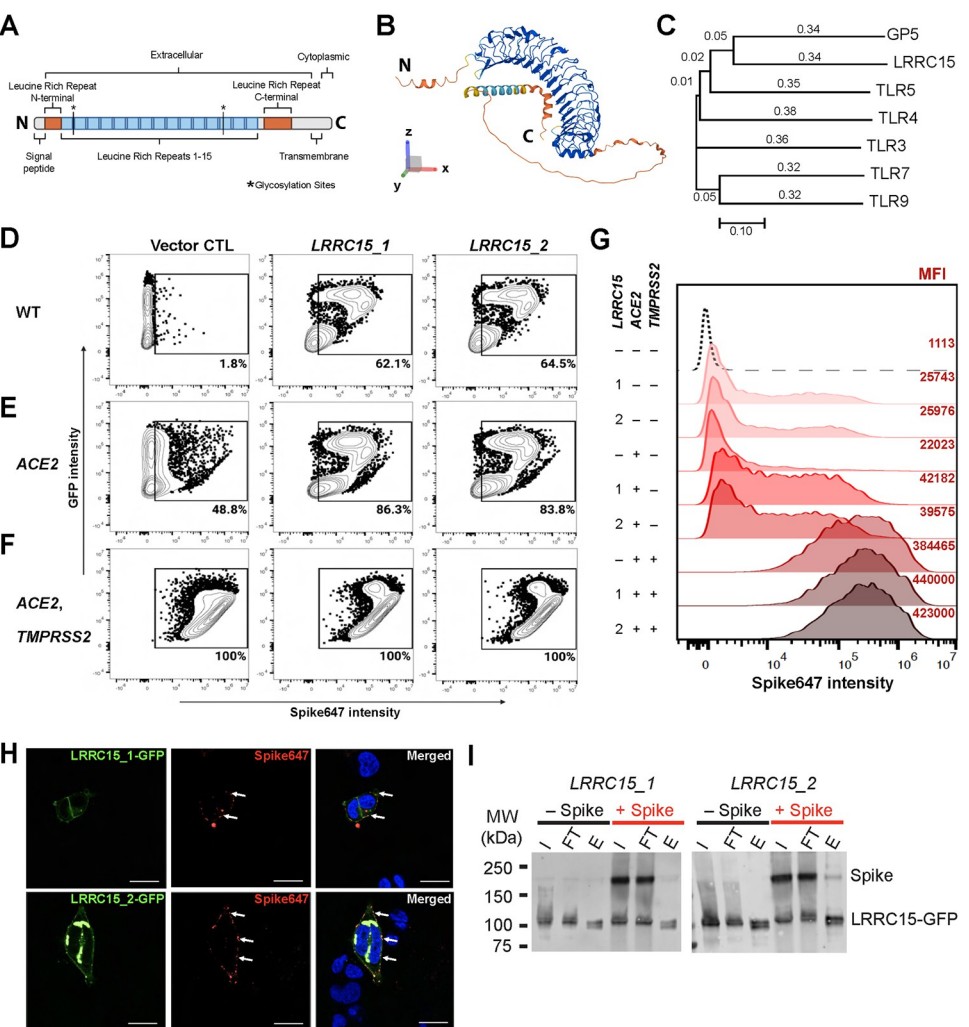

**Fig 3. Confirmation that LRRC15 binds to SARS-CoV-2 spike protein. (A)** LRRC15 contains 15 leucine-rich repeats, a short cytoplasmic C-terminus, and 2 glycosylation sites. **(B)** Predicted protein structure of LRRC15 (from alpha fold). **(C)** LRRC15 is part of the LRR-Tollkin family. **(D)** Flow cytometry analysis of Alexa Fluor-647 (Spike647) binding in WT HEK293T cells, **(E)** HEK293T-*ACE2*, and **(F)** HEK293T cells with stable expression of *ACE2* cDNA and *TMPRSS2* cDNA (HEK293T-*ACE2-TMPRSS2*). Each cell line was transfected with plasmids encoding cDNA for *GFP*-tagged *LRRC15* (transcript 1 or 2) or with empty *GFP* vector as negative control plasmid. **(G)**. Histogram summary shows MFI of **(D–F)**. **(H)** Representative images of interaction between LRRC15-GFP and Alexa Fluor 647-conjugated SARS-CoV-2 HexaPro spike protein in HEK293T cells ($N = 2$). Images were taken at 40× magnification. Green = LRRC15-GFP, red = Spike647, blue = Hoechst-stained nuclei. Scale bar = 25 $\mu$m. **(I)** Immunoprecipitation of LRRC15 with spike protein. Lysates of HEK293T cells transfected with GFP-tagged *LRRC15* (transcript 1 or 2, *LRRC15_1* and *LRRC15_2*, respectively) incubated with SARS-CoV-2 HexaPro spike protein were immunoprecipitated using anti-LRRC15 primary antibody. Immunoblots were performed for LRRC15 and for SARS-CoV-2 HexaPro spike. I = input, FT = flow-through, E = elute. The data underlying all panels in this figure can be found in DOI: 10.5281/zenodo.7416876. ACE2, angiotensin-converting enzyme 2; GP5, glycoprotein V platelet; LRRC15, leucine-rich repeat-containing protein 15; MFI, mean fluorescence intensity; SARS-CoV-2, Severe Acute Respiratory Syndrome Coronavirus 2; TLR, toll-like receptor; TMPRRS2, Transmembrane serine protease 2.

in infection with authentic SARS-CoV-2 virus. HEK293T-*ACE2-TMPRSS2* cells were infected with increasing doses of SARS-CoV-2 (Wuhan variant, **S4E Fig**) and cell death was assessed 48 h later. Ectopic expression of LRRC15 did not inhibit infection (two-way ANOVA, $p = 0.378$). Together, these data show that LRRC15 is not sufficient to confer SARS-CoV-2 tropism. Instead, LRRC15 can limit SARS-CoV-2 spike-mediated entry in *cis*, but once replication competent virions have entered cells, LRRC15 cannot protect infected cells from death.

## LRRC15 is found on lung fibroblasts that are not infected by SARS-CoV-2

At the tissue level, *LRRC15* RNA is most abundant in the placenta, with expression also found in skin, tongue, tonsils, and lung [14]. At the single-cell level, we used the COVID-19 Cell Atlas dataset to confirm *LRRC15* expression in placenta decidua stromal cells [15], multiple lymphatic vessels [16–19], and fibroblasts from the skin [20], prostate [21], and lung [17,22–26] (**Fig 4A**). In the lung [26] (**Fig 4B**), we found *LRRC15* is primarily expressed in fibroblasts as well as a population annotated as "neuronal cells" (**Fig 4C**), and these populations were not infected with SARS-CoV-2 (**Fig 4D**). These data were corroborated by 2 other COVID-19 patient single cell/nucleus RNA seq datasets that show similar *LRRC15* fibroblast expression profiles (**S5A–S5F Fig**), and again LRRC15$^+$ cells were not infected with SARS-CoV-2 [24] (**S5C Fig**). Of note, the viral RNA detected was low in these patients. Together, these data support our *in vitro* observations that LRRC15 does not mediate SARS-CoV-2 infection but may instead act as an innate immune barrier. In contrast, *ACE2* was detected primarily in uninfected type I (AT1) and (AT2) alveolar epithelium (**Fig 4D**), and SARS-CoV-2-infected alveolar epithelium ("Other epithelial cells") that lost AT1/2 markers and up-regulated ribosomal transcripts consistent with viral infection and cell death. We next assessed LRRC15 protein expression in human lung parenchyma. We observed that COVID-19 lungs have epithelial metaplasia, more immune infiltrate, and intra-alveolar fibroblast proliferation (**Figs 4E and S6A**). This matches with single-cell RNAseq that shows lung fibroblasts increases significantly during COVID-19 (7.9% in control and 22.9% in COVID-19 patients; **Fig 4G**). Moreover, we found that LRRC15 is present on the alveolar surface of lung tissue samples from donors with COVID-19, but not present in control lungs from individuals without COVID-19, and LRRC15 expression was mutually exclusive with collagen (**Figs 4H and 4I and S6B**).

While HEK293T cells do not express *LRRC15*, the human fibroblast line IMR90 does (**Fig 5A**). In the rat glia cell line C6, *LRRC15* is mildly regulated in response to proinflammatory cytokines like IL1β, IL6, and TNF [27], and more recently TGFβ signaling has been linked to LRRC15 expression in cancer-associated fibroblasts [28,29]. In human fibroblasts, we found TGFβ up-regulates both *LRRC15* (**Fig 5C**) and *COL1A1* transcripts (**Fig 5D**) and LRRC15$^+$ fibroblasts bind SARS-CoV-2 spike protein (**S7A Fig**). Moreover, ectopic expression of LRRC15 is also sufficient to enhance SARS-CoV-2 spike binding on fibroblasts (**Fig 5D**); however, LRRC15 expression was again not sufficient to confer SARS-CoV-2 pseudovirus tropism (**Fig 5E**). Since human lung fibroblasts express LRRC15 and are not infected with SARS-CoV-2, we reasoned that LRRC15 may act to bind and sequester SARS-CoV-2 virions away from ACE2$^+$ target lung epithelium. In human COVID-19 patients, fibroblasts and epithelial cells are present at a ratio of approximately 2:1 (**Fig 5F**). Thus, to test if LRRC15 can sequester virus and suppress SARS-CoV-2 infection, we co-cultured fibroblasts expressing *LRRC15-GFP* with SARS-CoV-2 permissive HEK293T-*ACE2-TMPRSS2* at a ratio of 2:1. Indeed, we found LRRC15$^+$ fibroblasts can antagonize infection of both SARS-CoV-2 pseudovirus (**Fig 5G**) and authentic SARS-CoV-2 virus (Wuhan, **Fig 5H**). Thus, together we show LRRC15 is expressed specifically by lung fibroblasts, is found coating the airways in COVID-19 patients, and mechanistically, LRRC15 can act to sequester SASR-CoV-2 virus and help suppress infection, which may potentially help protect ACE2$^+$ alveolar epithelium in patients with COVID-19.

## LRRC15 is a potent regulator of antiviral and fibrotic programs

While we believe the most direct mechanism by which LRRC15 may participate in COVID-19 infection is through binding to and sequestering SARS-CoV-2 virions, little is known about the broader role for LRRC15 in physiology or how LRRC15 expression impacts fibroblast transcriptional programs. A recent study on the organization of tissue fibroblasts identified

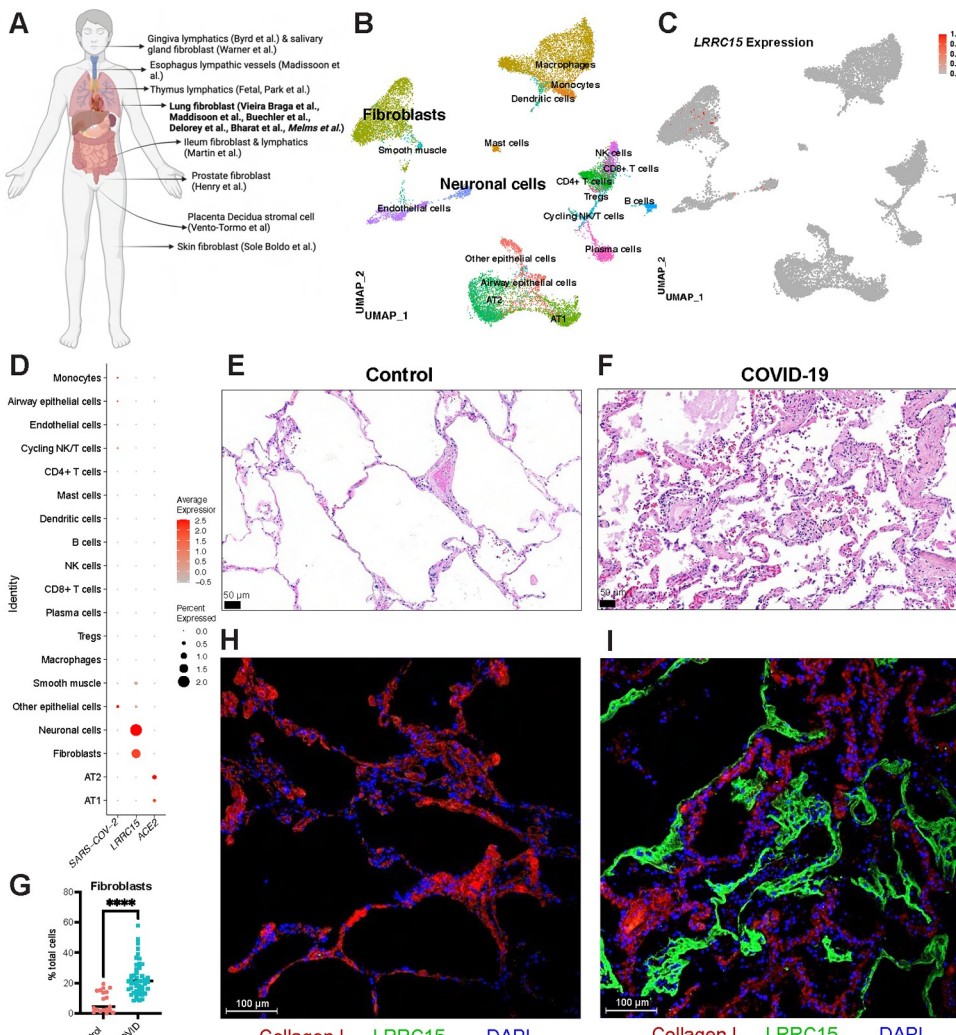

**Fig 4. LRRC15 is expressed in lung fibroblasts and lines the airways in COVID-19 patients. (A)** Overview of cell types expressing *LRRC15* from existing single-cell RNA-sequencing datasets. **(B)** UMAP plot of lung single-nucleus RNA seq dataset (Melms and colleagues). **(C)** Feature plot and **(D)** dotplot shows *LRRC15* is expressed in fibroblasts and neuronal cells. Expression of *LRRC15* is also observed in fibroblasts of separate studies (See **S5 Fig**). **(E)** Proportion of cells that are lung fibroblasts increases with COVID lungs (7.9% in control ($N$ = 19) and 22.9% in COVID ($N$ = 47); unpaired $t$ test, $p < 0.0001$). **(F and G)** Representative micrograph of HE-stained lung tissue section obtained from **(F)** a human donor without COVID-19 and **(G)** donor diagnosed with COVID-19. Imaging performed at 200× magnification (scale bar = 50 m). All images in **S6A Fig** (Control, $N$ = 1; COVID-19, $N$ = 4). **(H and I)** Representative micrograph of immunofluorescence staining in human lung tissue section obtained from **(H)** donor without COVID-19 and **(I)** donor diagnosed with COVID-19. Images were taken at 200× magnification (scale bar = 100 m). Red = Collagen I, green = LRRC15, blue = DAPI. All images in **S6B Fig** (Control, $N$ = 3; COVID-19, $N$ = 4). The data underlying all panels in this figure can be found in DOI: 10.5281/zenodo.7416876. ACE2, angiotensin-converting enzyme 2; COVID-19, Coronavirus Disease 2019; HE, hematoxylin and eosin; LRRC15, leucine-rich repeat-containing protein 15.

*LRRC15* as a lineage marker for perturbed state activated myofibroblasts [23]. These specialized fibroblasts arise during disease, express collagen and other ECM-modifying genes, and participate in tissue repair and fibrosis [23]. We also observed lung *LRRC15*+ myofibroblasts in multiple COVID-19 patient datasets, and these cells also express collagen (**Fig 6A**). To directly investigate the relationship between LRRC15 and collagen, we generated an LRRC15 overexpressing stable human fibroblast line, then evaluated LRRC15-induced transcriptional

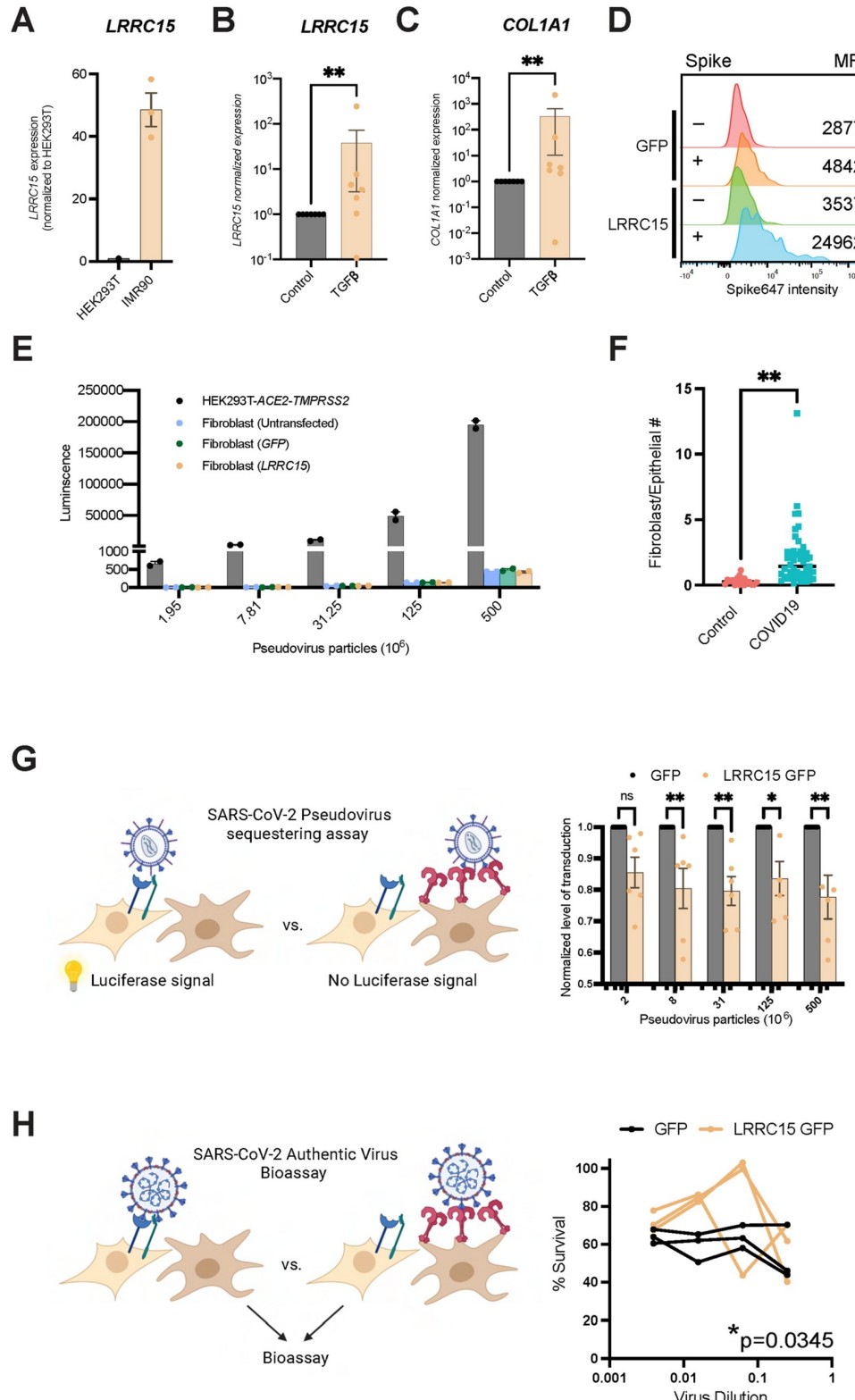

**Fig 5. LRRC15 is not a SARS-CoV-2 entry receptor but inhibits infection in *trans*. (A)** IMR90 fibroblasts express *LRRC15*, quantified via RT-qPCR. $N = 3$ per cell line. **(B and C)** TGFβ increased **(B)** *LRRC15* and **(C)** *COL1A1* in fibroblasts, quantified via RT-qPCR. $N = 7$ for each group. Significance was determined by Mann–Whitney one-tailed test, $**p < 0.01$. **(D)** IMR90 fibroblasts expressing *LRRC15* bind spike, MFI = mean fluorescence intensity. **(E)**

Fibroblasts do not have innate tropism for SARS-CoV-2 and overexpression of *LRRC15* does not mediate infection. Transduction efficiency (luciferase luminescence) was compared to permissive cell line HEK293T-*ACE2-TMPRSS2*. $N = 2$ independent replicates for each group. **(F)** Pooled analysis of 3 independent studies indicate ratio of fibroblasts to epithelial cells in COVID-19 lungs is approx. 2:1 (0.3 in control ($n = 19$) and 2.06 in COVID-19 ($n = 47$); unpaired two-tailed t test, $p < 0.0001$). **(G)** LRRC15 expressing fibroblasts can suppress SARS-CoV-2 spike pseudovirus infection of HEK293T-*ACE2-TMPRSS2* cells. Significance was determined by two-way ANOVA, Sidak's multiple comparison test, $^{**}p < 0.01, ^*p < 0.05$. $N = 6$ per condition. **(H)** LRRC15 expressing fibroblasts can suppress authentic SARS-CoV-2 infection of HEK293T-*ACE2-TMPRSS2* cells. Significance was determined by two-way ANOVA, Sidak's multiple comparison test, $^*p < 0.05$. $N = 3$ per condition. The data underlying all panels in this figure can be found in DOI: 10.5281/zenodo.7416876. ACE2, angiotensin-converting enzyme 2; COL1A1, collagen type I alpha 1 chain; LRRC15, leucine-rich repeat-containing protein 15; SARS-CoV-2, Severe Acute Respiratory Syndrome Coronavirus 2; TMPRRS2, Transmembrane serine protease 2.

changes by RNAseq. Surprisingly, we found driving expression of LRRC15 in fibroblasts induced up-regulation of cellular antiviral programs and down-regulated expression of collagen transcripts (**Fig 6B** and **S5 Table**). These results are clearly visible in the volcano plot (red transcripts were up-regulated, blue transcripts down-regulated); however, they were also captured in pathway analysis (**Fig 6C** and **S6 Table**), where we found interferon and influenza signaling were the most up-regulated pathways, whereas wound healing and pulmonary fibrosis were the most down-regulated pathways. Of note, pancreatic adenocarcinoma signaling was also highly up-regulated, and this is in line with a recent study highlighting the role of LRRC15[+] fibroblasts in driving disease severity in pancreatic cancer [29]. The primary antiviral pathways up-regulated by LRRC15 expression were IFITs (interferon-induced proteins with tetratricopeptide repeats), MXs (Myxovirus resistance genes), and OASs (2-prime, 5-prime oligoadenylate synthetases) and we confirmed these data by RT-qPCR (**Fig 6D**). Moreover, LRRC15 expression had an unexpected and potent ability to down-regulate collagen transcripts, and we confirmed this both by RT-qPCR (**Fig 6E**) and western blotting (**Figs 6F** and **S6B).**

Overall, we describe the TLR-related receptor LRRC15 as a new spike receptor that can bind and sequester SARS-CoV-2 and limit infection. LRRC15 is induced extensively during COVID-19, where it lines the airways and may form an innate antiviral barrier. Surprisingly, while LRRC15 is induced on fibroblasts during disease, ectopic expression of LRRC15 switches fibroblast transcriptional programs from a fibrotic program to an antiviral one, and this may help the lung orchestrate innate immunity programs versus immune resolution and lung repair.

## Discussion

Using an unbiased functional genomics approach, we have identified the leucine-rich repeat receptor LRRC15 as a new SARS-CoV-2 inhibitory receptor that can regulate innate immunity and lung repair. LRRC15 promotes SARS-CoV-2 spike binding comparable to ACE2; however, LRRC15 is not sufficient to confer viral tropism. LRRC15 is normally highly expressed in tissues that form important immune barriers like the placenta, skin, and various lymphatics, and is related to TLR innate immune receptors [14]. In previous work, LRRC15 has been shown to suppress adenovirus infection [30], and here, we show LRRC15 can also bind to and suppress SARS-CoV-2 spike pseudovirus and live SARS-CoV-2 infection. Moreover, in human SARS-CoV-2-infected airways, we see that LRRC15 forms a pronounced barrier-like structure, and given the expression pattern and function of LRRC15, we hypothesize that this molecule is a pattern recognition receptor and innate immune barrier that may play an important role in host defense. Moreover, LRRC15 is found on collagen-producing myofibroblasts, and we show ectopic expression of LRRC15 suppresses collagen production and drives

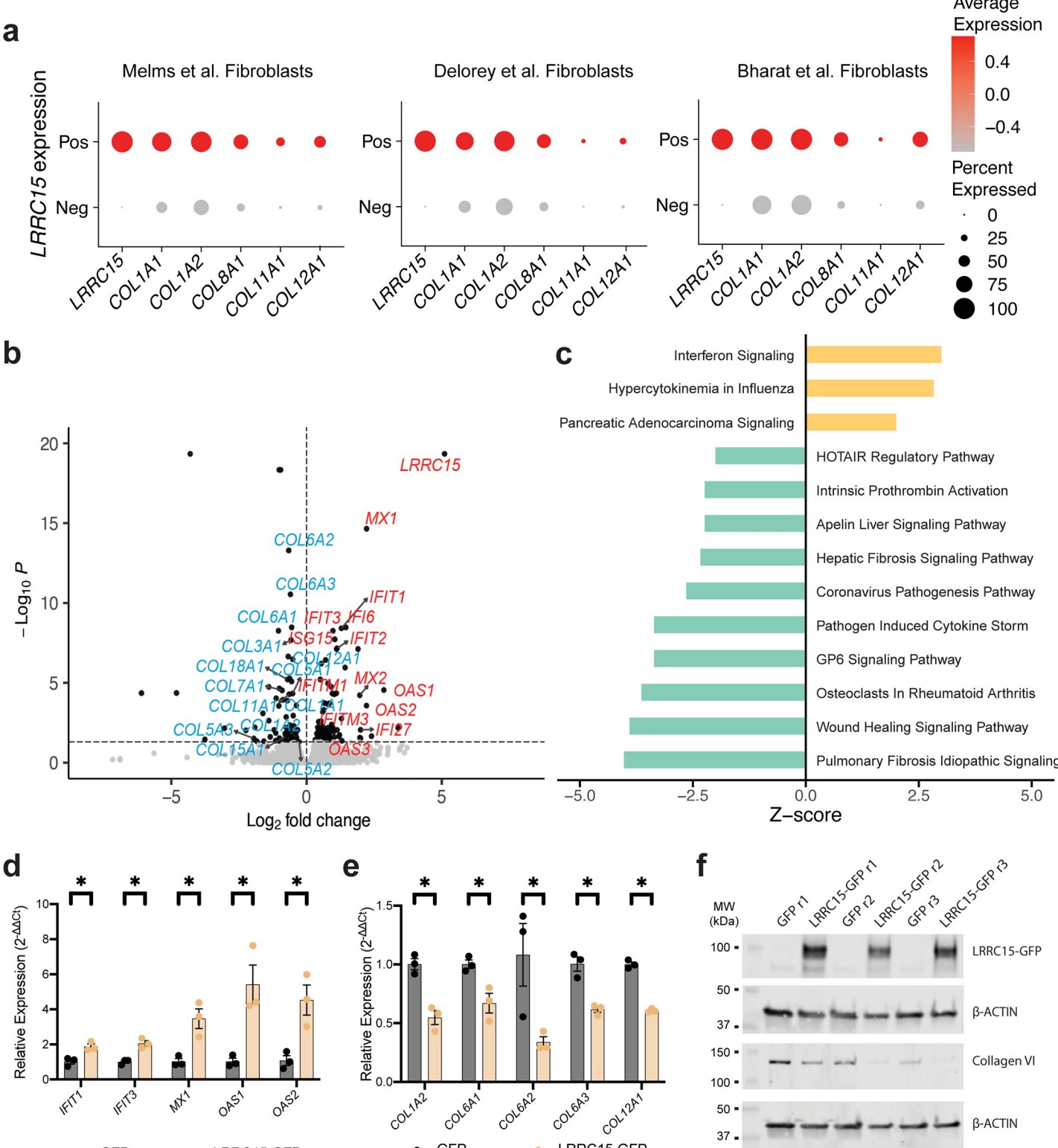

**Fig 6. LRRC15 expression is correlated with collagen and antiviral gene signatures.** (**A**) *LRRC15*⁺ fibroblasts have an enhanced collagen gene signature. Dotplots generated from 3 separate studies. Pos = *LRRC15*⁺, Neg = *LRRC15*⁻. (**B**) Volcano plot of DEGs from fibroblasts ectopically expressing either LRRC15 or GFP. *N* = 3 biological replicates for each group. A subset of DEGs is labeled, including collagen genes and genes related to antiviral signaling. Blue labels indicate down-regulation, while red labels indicate up-regulation. (**C**) DEG-associated canonical pathways as determined by Ingenuity Pathway Analysis. Canonical pathways were filtered to show

only those with $p$-value $< 0.05$ and with a nonzero Z-score. **(D and E)** *LRRC15* overexpression causes **(D)** up-regulation of antiviral transcripts and **(E)** down-regulation of collagen transcripts by RT-qPCR. Results were calculated using the $\Delta\Delta C_T$ method, normalized to the average of control *GFP*-only cells. Significance was assessed using one-tailed Mann–Whitney test, $^*p < 0.05$. $N = 3$ per condition. **(F)** Ectopic expression of LRRC15 in fibroblasts decreases Collagen VI protein expression compared to *GFP*-only control cells. Western blots for LRRC15 and Collagen VI. The data underlying all panels in this figure can be found in DOI: 10.5281/zenodo. 7416876. DEG, differentially expressed gene; IFIT, interferon-induced proteins with tetratricopeptide repeats; LRRC15, leucine-rich repeat-containing protein 15 MX, Myxovirus resistance family; OAS, 2-prime, 5-prime oligoadenylate synthetase family.

antiviral programs, and in this way, directly links SARS-CoV-2 with innate antiviral immunity and lung fibrosis.

Although our data shows that LRRC15 promotes cellular binding to SARS-CoV-2 spike protein, we also show that LRRC15 does not act as an entry receptor, but instead can inhibit SARS-CoV-2 in *trans*. This observation is consistent with a previous report that LRRC15 can also impede adenovirus infection [30]. We hypothesize that LRRC15 may play a role in limiting SARS-CoV-2 transmission by sequestering free virus in the airways of COVID-19 patients, and the LRRC15 we observed lining the airways may also suppress collagen deposition protecting the airways from fibrosis during some stages of lung infection. It is likely that the role for LRRC15 in lung immunity is broader than just interactions with SARS-CoV-2, and LRRC15 may represent a new fibroblast-expressed pattern recognition receptor that can bind to and sequester a variety of microbial antigens.

LRRC15 is a member of the LRR superfamily and LRR-Tollkin subfamily of LRR-containing proteins, many of which play critical roles in host defense [12]. Of the TLR family, LRRC15 is most related to TLR5, which also recognizes a major extracellular virulence factor, the bacterial extracellular protein flagellin [31]. Remarkably, while our manuscript was in preparation and then review, 2 other groups independently released preprints describing similar ORF/CRISPR activation screening strategies to identify new host factors that can regulate spike binding; both screens also pulled out LRRC15 as a top factor driving spike/host cell interactions [32,33]. These studies corroborate our findings, despite their use of different spike formulations, overexpression strategies, and cell lines. Moreover, since our initial submission [34], Song and colleagues have replicated our finding that LRRC15 can act in *trans* to suppress SARS-CoV-2 infections and this has now been published [33]. Together, our studies highlight a fundamental new role for LRRC15 in SARS-CoV-2 biology and likely beyond.

Several CRISPR Loss of Function (LOF) and Gain of Function (GOF) screens have been reported in attempts to identify novel SARS-CoV-2 interactors and regulators. Though these CRISPR screens have been successful in identifying novel SARS-CoV-2 receptors and co-receptors [35–39], ACE2-regulators [40,41], complexes such as the vacuolar ATPase proton pump, Retromer, Commander and SWI/SNF chromatin remodeling machinery [40,41] and have implicated many new pathways in SARS-CoV-2 infection [36,38,41], they have all failed to identify LRRC15. This difference is likely due to screening with SARS-CoV-2 authentic virus and pseudovirus screens being unable to divorce spike binding from downstream effects of infection. Our fluorophore-conjugated spike protein/pooled CRISPR screening model thus represents a new and complementary paradigm for investigating host/virus interactions or virtually any other cell surface interaction.

Our data suggests the primary mechanism of action for LRRC15 in the context of SARS-CoV-2 infection is likely through a direct interaction with the spike protein that sequesters SARS-CoV-2 virions and in this way helps to limit infection. Beyond this, we show that LRRC15 also has a potent and specific impact on fibroblast gene expression, suppressing collagen while enhancing antiviral programs. We found LRRC15 expression caused an up-regulation of 3 antiviral pathways, IFIT, MX, and OAS, and these antiviral pathways are also up-regulated in primates infected with SARS-CoV-2 [42] and COVID-19 patients [43]. IFIT

proteins are induced by IFN, viral infection, or PAMP recognition, where they can then directly bind to viral RNA, block viral translation, and activate cellular antiviral responses [44]. A recent preprint found the SARS-CoV-2 nonstructural protein NSP16 helps SARS-CoV-2 evade the host antiviral immune response by avoiding the antiviral activities of IFIT1 and 3 [45]. MX proteins are interferon-induced dynamin-like GTPases with antiviral activity against multiple RNA and DNA viruses. For example, MX1 can block influenza A by altering sorting of viral vesicles in the ER/Golgi intermediate compartment [46]. In COVID-19, MX1 is up-regulated with increasing viral load [47], and functionally, the SARS-CoV-2 protein ORF6 can suppress MX1 induction [48]. OAS proteins are dsRNA sensors that can activate RNAse L which then degrades viral RNA and inhibits protein synthesis [49]. Importantly, OAS1 is a potent host antiviral factor that can block SARS-CoV-2 infection in vitro, and OAS1 expression also associates with protection from severe COVID-19 outcome in vivo [50].

While our data highlights a new role for LRRC15 in promoting SARS-CoV-2 spike binding, limiting infection, and regulating collagen expression, it is currently unclear how LRRC15 contributes to human COVID-19 disease. Notably, while this manuscript was under revision, a preprint authored by Gisby and colleagues investigated serum from control and COVID-19 infected end stage kidney disease patients, and found that out of the entire serum proteome, depletion of circulating LRRC15 is the strongest predictor of COVID-19 clinical outcome [51]. Integrating these data, it is possible that fibroblast-expressed LRRC15, or potentially cell free LRRC15 deposits in the airways, could trap viral particles for subsequent clearance by the innate immune system, while at the same time enhancing cellular antiviral tone and suppressing fibrosis. LRRC15 may even help fibroblasts pass immobilized virus to innate lung antigen presenting cells, as a recent spatial transcriptomics study showed that lung fibroblasts interact with both SARS-CoV-2 Spike+ macrophages and dendritic cells [52]. If lung LRRC15/SARS-CoV-2 complexes are depleted, and new LRRC15 is not produced, this may lead to a detectable decrease in serum LRRC15 that is indicative of poor COVID-19 outcome.

Overall, our unbiased functional genomic investigation of SARS-CoV-2 spike/host interactions identified the novel TLR-related receptor LRRC15 as a powerful host factor driving SARS-CoV-2 spike interactions and controlling both antiviral and antifibrotic responses. Further investigation into how LRRC15 contributes to innate immunity can help us better understand and treat this and future pandemics.

## Experimental model and subject details

**Cell culture.**   HEK293T cells (female; ATCC, CRL-3216, RRID: CVCL_0063) were cultured in Dulbecco's Modified Eagle Medium (Thermo Fisher Scientific, Cat #11995065) with 10% HyClone Fetal Bovine Serum (Cytiva, SH30084.03) and 1% Penicillin-Streptomycin (Gibco, 15140122) at 37°C, 5% $CO_2$ and atmospheric oxygen. IMR90 *E6E7* (female) human fibroblast cells were a gift from Anthony Cesare (Children's Medical Research Institute, Sydney, Australia). IMR90 were cultured in DMEM (Thermo Fisher Scientific, 11995065) supplemented with 10% HyClone FBS (Cytiva, SH30084.03) and 1× non-essential amino acids (Gibco, 11140050) at 37°C, 3% $O_2$ and 10% $CO_2$. Expi293F cells (female; Thermo Fisher Scientific, A14527, RRID:CVCL_D615) were cultured in Expi293 Expression Medium (Thermo Fisher Scientific, A1435101) with 5% $CO_2$ and atmospheric $O_2$ at 37°C for 24 h and then lowered to 32°C for 72 h. Cell lines have been authenticated.

## Method details

**Generation of CRISPR activation cell line.**   HEK293T cells were co-transfected with pPB-R1R2_EF1aVP64dCas9VP64_T2A_MS2p65HSF1-IRESbsdpA (Addgene #113341) and

the Super PiggyBac Transposase Expression Vector (System Biosciences, PB210PA-1) using Lipofectamine 3000 Transfection Reagent (Thermo Fisher Scientific). These cells (HEK293T--CRISPRa) were then selected on blasticidin (Merck) at 5 μg/mL for 10 days prior to clonal isolation and expansion. These cells express synergistic activation machinery (SAM), which includes VP64-dCas9-VP64 fusion protein and helper proteins MS2, p65, and HSF. When transduced with pXPR_502 (Addgene #96923) sgRNA plasmid, the cells also express PCP-p65-HSF complex that is recruited to PP7 aptamers in the sgRNA scaffold [5].

**sgRNA vector cloning.** Single guide RNA (sgRNA) sequences for non-targeting control (NTC) and ACE2 were taken from the Weissman Human Genome-wide CRISPRa-v2 library (Addgene #83978). LRRC15 sgRNA sequences and additional ACE2 sgRNA sequences were taken from the Human CRISPR activation pooled library set A (Addgene #92379). Sense and antisense strands for each sequence were ordered as DNA oligonucleotides (IDT) with 5′ overhangs of 5′-CACC-3′ on the sense strand oligonucleotide and 5′-AAAC-3′ on the antisense strand oligonucleotide. Oligonucleotides were annealed at 4°C for 16 h and pXPR-502 (Addgene #96923) was digested with Esp3I (Thermo Fisher Scientific, ER0451) or BsmBI-v2 (New England Biolabs). sgRNA DNA oligonucleotide duplexes were ligated into the digested pXPR-502 backbone using T4 ligase (New England Biolabs) and incubated at 4°C overnight. NEB 10-beta competent *E. coli* (New England Biolabs) were transformed with 100 ng of each sgRNA construct by heat-shock, plated onto LB-agar plates (Life Technologies) containing ampicillin (Sigma-Aldrich) and grown at 37°C. Individual colonies were picked, expanded in Luria broth (Life Technologies) supplemented with ampicillin and amplified constructs were harvested using either ISOLATE II Plasmid Mini Kit (Bioline) or PureYield Plasmid Maxiprep System (Promega Corporation).

**Whole-genome sgRNA library amplification.** MegaX DH10B T1$^R$ Electrocomp Cells (Thermo Fisher Scientific) were electroporated with 400 ng Human CRISPR activation pooled library set A (Addgene #92379) and left to recover in Recovery Medium for 1 h at 37°C. Cells were then spread on 600 cm$^2$ LB-agar plates supplemented with carbenicillin (Merck) and incubated at 37°C for 16 h. All colonies were scraped, collected, and processed using the Pure-Yield Plasmid Maxiprep System (Promega Corporation). The concentration of the plasmid library was determined via Nanodrop (Thermo Fisher Scientific).

**Lentivirus production and viral transduction.** Lipofectamine 3000 Transfection Reagent (Thermo Fisher Scientific) in Opti-MEM Medium (Gibco) was used to co-transfect HEK293T cells with psPAX2 (Addgene #12260), pCAG-VSVg (Addgene #35616), and either individual sgRNA constructs ligated into pXPR-502 (Addgene #96923) or pooled CRISPRa library (Human CRISPR activation pooled library set A, Addgene #92379) according to the manufacturer's instructions. Cells were incubated with transfection reagents for 16 h before the media was replaced. Viral media was collected 24 h later. For individual sgRNA constructs, neat viral media was added to HEK293T-CRISPRa cells with Polybrene Infection/Transfection Reagent (Sigma-Aldrich) at a concentration of 8 μg/mL. Viral media was replaced with fresh medium the following day and puromycin dihydrochloride (Gibco) added 24 h later at a concentration of 1.6 μg/mL for 72 h selection. For sgRNA library virus, viral media was passed through a 0.45 μm filter (Merck Millipore) and concentrated using 100K MWCO Pierce Protein Concentrators (Life Technologies Australia). Concentrated virus was then stored at −80°C.

**SARS-CoV-2 spike protein production.** The expression construct for recombinant soluble trimeric SARS-CoV-2 spike protein (residues 1–1208, complete ectodomain) was generously provided by Dr. Florian Krammer (Icahn School of Medicine, Mt. Sinai). This protein was used for the initial setup of the screen (shown in Fig 1) and in 1 CRISPRa screen (screen 2). This construct includes the SARS-CoV-2 spike native signal peptide (residues 1–14) to target the recombinant protein for secretion, stabilising proline substitutions at residues 986 and

987, substitution of the furin cleavage site (residues 682–685) with an inert GSAS sequence, and a C-terminal His6-tag to enable affinity purification.

Soluble trimeric SARS-CoV-2 spike was expressed in EXPI293F cells via transient transfection using 25 kDa linear polyethyleneimine (PEI) (Polysciences). EXPI293F cultures were grown at 37˚C, with shaking at 130 rpm, to a cell density of $3 \times 10^6$ cells/mL before transfection with pre-formed SARS-CoV-2 spike plasmid DNA:PEI complexes (2 μg/mL DNA and 8 μg/mL PEI). The transfected cells were incubated at 37˚C for 24 h and then at 32˚C for a further 72 h before harvesting. Culture medium, containing secreted SARS-CoV-2 spike, was harvested by centrifugation at 4,000 ×g for 20 min. Supernatants from the centrifugation step were supplemented with 20 mM HEPES (pH 8.0) and subjected to immobilized metal affinity chromatography (IMAC) by incubation with Ni-NTA agarose pre-equilibrated with a buffer consisting of 20 mM $NaH_2PO_4$ (pH 8.0), 500 mM NaCl, and 20 mM imidazole. His6-tagged SARS-CoV-2 spike protein was eluted from the Ni-NTA agarose using a buffer comprising 20 mM $NaH_2PO_4$ (pH 7.4), 300 mM NaCl, and 500 mM imidazole. Eluates from affinity chromatography were concentrated and further purified by gel filtration chromatography using a Superdex 200 10/30 GL column (Cytiva) and buffer consisting of 20 mM HEPES (pH 7.5) and 150 mM NaCl. The quality of protein purification was assessed by SDS-PAGE and multiple angle laser light scattering (MALLS).

The expression construct for a more stable variant of soluble trimeric SARS-CoV-2 spike ectodomain protein called "HexaPro" was a gift from Jason McLellan (Addgene, #154754). This "Hexapro" protein was used in 2 CRISPRa screens (screen 1 and 3) and in all validation experiments. This construct, in addition to above, includes 6 total stabilising proline substitutions at residues 817, 892, 899, 942, 986, and 987. The protein was expressed, and the culture medium was harvested as above. The supernatant containing the protein was supplemented with 20 mM HEPES (pH 8.0) and subjected to IMAC with Ni-NTA as above. The eluate was dialysed to a buffer containing 2 mM Tris (pH 8.0) and 200 mM NaCl and concentrated to reduce the total volume by a factor of 3. The sample was passed through a 0.22 μm filter and purified by gel filtration chromatography using HiLoad 16/600 Superdex 200 (Cytiva) in a buffer composed of 2 mM Tris (pH 8.0) and 200 mM NaCl. The quality of the protein was assessed by SDS-PAGE and MALLS.

**Conjugation of SARS-CoV-2 spike glycoprotein with fluorophores.** Spike protein was conjugated to Alexa Fluor 488 or Alexa Fluor 647 using protein labeling kits (Invitrogen) according to manufacturer's instructions. Briefly, 50 μL of 1 M sodium bicarbonate was added to 500 μl of 2 mg/mL spike protein. The solution was then added to room temperature Alexa Fluor 488 or 647 reactive dye and stirred for 1 h at room temperature. Conjugated spike proteins were loaded onto Bio-Rad BioGel P-30 fine size exclusion purification resin column and eluted via gravity (Alexa Fluor 488) or centrifugation (Alexa Fluor 647). NanoDrop (Thermo Fisher Scientific) was used to determine protein concentration.

**Generation of ACE2 and dual ACE2/TMPRSS2 cDNA overexpression cell lines.** HEK293T cells stably expressing human ACE2 (HEK293T-*ACE2*) were generated by transducing HEK293T cells with a lentivirus expressing *ACE2* [53]. Briefly, ACE2 ORF was cloned into a third-generation lentiviral expression vector, pRRLsinPPT.CMV.GFP.WPRE [54] using Age1/BsrG1 cut sites, thus replacing *GFP ORF* with *ACE2* to create a novel expression plasmid, herein referred to as ppt-*ACE2*. Lentiviral particles expressing ACE2 were produced by co-transfecting ppt-*ACE2*, a second-generation lentiviral packaging construct psPAX2 and VSV-G plasmid pMD2.G (Addgene #12259) in HEK293T cells by using polyethylenimine as previously described [55]. Virus supernatant was harvested 72 h post transfection, pre-cleared of cellular debris, and centrifuged at 28,000 ×g for 90 min at 4˚C to generate concentrated virus stocks. To transduce HEK293T cells, 10,000 cells per well were seeded in a 96-well tissue

culture plate and virus supernatant added in a 2-fold dilution series. At 72 h post transduction, the surface expression of ACE2 was measured by immunostaining the cells with anti-ACE2 monoclonal antibody (Thermo Fisher Scientific, MA5-32307). Cells showing maximal expression of ACE2 were then sorted into single cells using BD FACS Aria III cell sorter to generate clonal populations of HEK293T-*ACE2* cells.

For generating HEK293T cells expressing both ACE2 and TMPRSS2 (HEK293T-*ACE2-TMPRSS2*), HEK293T-*ACE2* cells described above were transduced with lentiviral particles expressing *TMPRSS2*. To achieve this, *hTMPRSS2a* (synthetic gene fragment; IDT) was cloned into lentiviral expression vector pLVX-IRES-ZsGreen (Clontech) using EcoR1/XhoI restriction sites and lentiviral particles expressing *TMPRSS2* were produced as described above. Lentiviral transductions were then performed on HEK293T-*ACE2* cells to generate HEK293T-*ACE2-TMPRSS2* cells. Clonal selection led to the identification of a highly permissive clone, HekAT24 [53], which was then used in subsequent experiments.

**Optimizing a flow cytometry-based assay for determining SARS-CoV-2 spike binding.** HEK293T-*ACE2* cells were dissociated by incubating with TrypLE for 5 min at 37°C and neutralized with DMEM. Approximately $10^6$ cells were collected, washed with 1% bovine serum albumin (BSA; Sigma-Aldrich) in Dulbecco's phosphate buffered saline (DPBS; Sigma-Aldrich), and then incubated with increasing concentrations of Alexa Fluor 488-conjugated SARS-CoV-2 spike glycoprotein (Spike488) for 30 min at 4°C. The cells were then washed once with DPBS before resuspending in 1% BSA in DPBS and analyzed using the Cytek Aurora (Cytek Biosciences). For cell mixing experiments, increasing proportions of HEK293T-*ACE2* cells (0%, 1%, 20%, 50%, 80%, and 100%) were combined with decreasing proportions of wild-type (WT) HEK293T cells (100%, 99%, 80%, 50%, 20%, 0%) to a total of $10^6$ cells per sample. These samples were incubated with 50 $\mu$g/mL Spike488 as described above and analyzed using the Cytek Aurora (Cytek Biosciences).

To confirm the validity of this assay in detecting binding in cells expressing CRISPRa machinery, a clonal line of HEK293T with stable expression of a plasmid encoding dCas9-VP64 and SAM system helper proteins (pPB-R1R2_EF1aVP64dCas9VP64_-T2A_MS2p65HSF1-IRESbsdpA) (HEK293T-CRISPRa) was transduced with lentivirus carrying *ACE2* sgRNA 1 (Weissman library) or NTC sgRNA. These cells were then incubated with Spike488 as previously described and analyzed on the Cytek Aurora (Cytek Biosciences).

**CRISPR activation screening.** HEK293T-CRISPRa cells were transduced with concentrated Human CRISPR activation pooled library set A (Addgene #92379)-carrying lentivirus at a multiplicity of infection (MOI) of approximately 0.3. Cells were selected on puromycin dihydrochloride (Gibco) at a concentration of 1.6 µg/mL for 3 days (screen 1 and 2); $3 \times 10^7$ cells (>500 cells/guide) were incubated with Spike488 for 30 min at 4°C, washed to remove excess spike protein, and sorted for increased Alexa Fluor 488 intensity using the BD FACSMelody Cell Sorter (BD Biosciences). Gates for flow assisted cytometric sorting were set using NTC sgRNA-transduced cells as a negative control and *ACE2* sgRNA-transduced cells as a positive control, both of which had been incubated with Spike488 under the same conditions as stated previously. Unsorted cells were maintained separately so as to be used as a diversity control. Cells were expanded and 1.5–2 × $10^6$ cells were then collected for gDNA extraction for sorted samples and $3 \times 10^7$ for the unsorted diversity control. Remaining diversity control cells were re-seeded and once again incubated with Spike488 under the same conditions as stated previously (screen 3). These spike-incubated cells were sorted again but selected on puromycin for 8 days prior to expansion and collection of $1 \times 10^7$ cells from both the sorted cell population and the unsorted diversity control population for gDNA extraction. Gating strategy is shown in **S1B Fig**.

gDNA was extracted from all collected cells using the ISOLATE II Genomic DNA Kit (Bioline). Samples were prepared for NGS via PCR. Genomic DNA (25 μg for unsorted diversity control samples, 5 μg for sorted samples) was added to NEBNext High-Fidelity 2X PCR Master Mix (New England Biolabs) and 0.4 μm P5 staggered primer mix and 0.4 μm of P7 indexing primer unique to each sample. PCR cycling conditions and primers were adapted from Sanson and colleagues [5]. Primer sequences can be found in **S3 Table**. Briefly, reactions were held at 95°C for 1 min, followed by 28 cycles of 94°C for 30 s, 53°C for 30 s, and 72°C for 30 s, followed by a final 72°C extension step for 10 min. Amplicons were gel extracted and purified using the ISOLATE II PCR & Gel Kit (Bioline) and the quality and concentration of DNA assessed with the High Sensitivity DNA kit (Agilent Technologies). Samples were then sent to Novogene for next-generation sequencing. Raw next-generation sequencing reads were then processed using MAGeCK (v0.5.9.2) [7] to identify enriched genes. Median normalization was used with gene test FDR threshold set to 0.1. Plots were generated using MAGeCKFlute (v1.12.0) [8] Normalized read counts were produced using MAGeCK "count" function on each pairing of unsorted diversity control and sorted sample. Mean and standard deviation was calculated for each individual sample (i.e., separately for diversity control and sorted sample) and the Z-score calculated using $Z = \frac{x-\mu}{\sigma}$, where x is the normalized read count for an individual sgRNA, μ is the mean of all normalized read counts in the sample, and σ is the standard deviation of all normalized read counts in the sample.

**Validation of ACE2 and LRRC15 by CRISPRa.** To validate the function of LRRC15 in binding SARS-CoV-2 spike, clonal HEK293T-CRISPRa cells were transduced with lentivirus carrying *ACE2* sgRNAs, *LRRC15* sgRNAs, or an NTC sgRNA. Cells were selected on 1.6 μg/mL puromycin dihydrochloride (Gibco) for 3 days and then collected for analysis by RT-qPCR and flow cytometry. For validation by flow cytometry, $1 \times 10^6$ cells were incubated with 50 *μg/mL* Spike647 as previously described and then analyzed using the Cytek Aurora (Cytek Biosciences). Binding affinity of ACE2 and LRRC15 were conducted with *ACE2* sgRNA3 and *LRRC15* sgRNA1 cells with 1, 5, 10, 25, 50, and 100 *μg/mL* Spike647 (corresponding to 7, 35, 70, 175, 350, and 700 nM).

**RNA extraction and RT-qPCR.** RNA was isolated from cells using the ISOLATE II RNA Mini Kit (Bioline) and concentration was measured by Nanodrop (Thermo Scientific). cDNA was synthesized using the iScript Select cDNA Synthesis Kit (Bio-Rad) according to manufacturer's instructions. Briefly, 50 to 500 ng of RNA was added to iScript RT Supermix and nuclease-free water to a final volume of 10 μL. The assembled reactions were then incubated in a thermocycler as follows: 25°C for 5 min, 46°C for 20 min, and then 95°C for 1 min. RT-qPCR was then performed on the cDNA samples using SYBR Select Master Mix (Thermo Fisher Scientific) and the LightCycler 480 System (Roche). All primer sequences used are listed in **S4 Table**. Results were analyzed using the $\Delta\Delta C_T$ method.

**LRRC15 structure prediction.** The predicted structure for LRRC15 was calculated using AlphaFold (v2.0) [56] (https://alphafold.ebi.ac.uk/entry/Q8TF66) and sourced via UniProt [57] (https://www.uniprot.org/uniprot/Q8TF66).

**LRR Tollkin phylogenetic tree.** Protein sequences of LRR Tollkin family members [12] were clustered using Clustal Omega (v1.2.2) [58]. The phylogenetic (Newick) tree was visualized with MEGA11 [58,59].

**Validation of LRRC15 independent of CRISPR activation.** *LRRC15-TurboGFP* fusion constructs (Origene, RG225990 and RG221437) were used for flow cytometry, immunoprecipitation, and immunocytochemistry experiments while *LRRC15-myc-DDK* fusion constructs (Origene, RC225990 and RC221437) were utilized for SARS-CoV-2 authentic virus inhibition experiments. Both *TurboGFP*-tagged and *Myc-DDK*-tagged *LRRC15* constructs were used in pseudovirus infection experiments assessing *cis*-inhibition of infection. *LRRC15* transcripts

were excised from the *LRRC15-TurboGFP* and *LRRC15-myc-DDK* constructs and replaced with multiple cloning sites to generate empty vector controls for transfection.

To evaluate the role of LRRC15 in binding SARS-CoV-2 spike glycoprotein independent of CRISPR activation machinery, 2.5 μg of plasmids carrying the *GFP*-tagged *LRRC15* cDNA transcript 1 or 2, or empty vector control (pLJM1-EGFP; Addgene #19319) were transfected into HEK293T, HEK293T-*ACE2*, and HEK293T-*ACE2-TMPRSS2* cells as described above. For each sample, $10^6$ cells were collected and incubated with Alexa Fluor 647-conjugated SARS-CoV-2 spike glycoprotein (Spike647) and analyzed using the Cytek Aurora (Cytek Biosciences) as described above.

**Immunoprecipitation.** For SARS-CoV-2 spike pulldown, $2 \times 10^7$ HEK293T cells transfected with *LRRC15-TurboGFP* (transcript 1 and 2) or *pLJM1-EGFP* (Addgene #19319) were incubated with 50 μg/mL HexaPro spike for 30 min at 4°C with rotation. Cells were washed with DPBS (Sigma-Aldrich, D8537) and incubated for 15 min in lysis buffer (1% Igepal-CA-630, 5 mM Tris HCl (pH 7.4), 150 mM NaCl, 1 mM $MgCl_2$, 5% glycerol, 10 mM NaF, 10 mM sodium pyrophosphate, 10 mM sodium orthovanadate, 60 mM β-Glycerophosphate, 1× complete EDTA-free protease inhibitor (Roche)) on ice. Samples were then sonicated at 90% amplitude for 30 s using the BANDELIN SONOPULS mini20 and spun down at 18,000 g for 10 mins. Concentration of protein samples was determined using BCA assay (Thermo Fisher Scientific). A total of 1 μg of anti-LRRC15 antibody (Abcam, EPR8188 (2)) or rabbit IgG (Covance, CTL-4112) was added to 1 mg protein lysate and incubated at 4°C with rotation for 2.5 h before precipitation with protein G (Thermo Fisher Scientific). Immunoprecipitated proteins were eluted with 0.1 M Tris and 4% SDC. Input, flow-through and eluate were mixed with 4× loading buffer and heated at 95°C for 5 min. Samples were loaded into pre-cast polyacrylamide gels (4% to 20% gradient, Bio-Rad) and electrophoresed at 90 V for 1.5 h. Proteins were transferred to 0.45 μm nitrocellulose membranes at 100 V for 1 h. Membranes were blocked in Intercept blocking buffer (LI-COR) for 30 min at room temperature with gentle agitation. Blocking solution was replaced with primary antibody (spike, LRRC15) Intercept buffer and membranes incubated overnight at 4°C with gentle agitation. Membranes were washed 3 times with TBST for 5 min with agitation prior to the incubation of membranes with secondary antibody in Intercept buffer for 2 h at room temperature. Membranes were washed another 3 times with TBST and then imaged using the Odyssey CLx (LICOR).

**Confocal imaging of cultured cells.** The 13 mm round coverslips were coated with Matrigel (Corning) diluted in DPBS and incubated for 30 min at 37°C. HEK293T cells transfected with *LRRC15* cDNA constructs were seeded onto the Matrigel-coated coverslips at a density of 50,000 cells per coverslip. The following day, cells were incubated with Alexa Fluor 647-conjugated SARS-CoV-2 spike protein at a concentration of 10 μg/mL in culture media for 30 min at 37°C. The cells were fixed in 4% paraformaldehyde (PFA) for 20 min at room temperature, washed 3 times with DPBS. Cells were incubated with Hoechst (1:2,000 in DPBS) for 20 min, washed 3 times, and mounted onto Superfrost plus slides (Fisherbrand) and then imaged using the Leica TCS SP8 STED 3× at 40× magnification.

**Patients.** Post-mortem formalin-fixed paraffin embedded (FFPE) tissue samples were obtained from 4 patients who died from severe COVID-19 infection and diagnosed using a PCR test between December 2020 and March 2021. The control group consisted of post-mortem FFPE lung tissue samples from 3 patients with melanoma metastases in the lung. Lung tissue from COVID-19 patients was compared with non-tumor lung tissue from melanoma patients. The study was approved by the Ethics Committee of Eastern Switzerland (BASEC Nr. 2020-01006/EKOS 20/71, BASEC Nr. 2016-00998/EKOS 16/015, and BASEC Nr. 2021-00678/EKOS 21/049).

**Hematoxylin and eosin (HE) staining, LRRC15 and collagen I immunofluorescence (IF).** Post-mortem lung tissue samples were collected and processed for paraffin embedding

according to standard diagnostic protocols in the Institute for Pathology of St. Gallen Cantonal Hospital. All tissue samples were routinely stained for histopathological diagnosis with HE following a standardized validated protocol. Two-micron-thick sections were cut using a Leica RM2255 rotary microtome (Leica Microsystems, DE) and placed on poly-L-lysine-coated slides. Slides were dewaxed in xylene, rehydrated, and subjected to heat-induced epitope retrieval in a sodium citrate solution (pH 6) for 20 min in a microwave oven. Slides were then allowed to cool to room temperature followed by a 60-min incubation with 1× PBS/5% skim milk at RT. Excess liquid was removed and sections were incubated for 18 h at 4°C with a polyclonal rabbit anti-human LRRC15 antibody (LSBio, catalog number LS-C405127, lot ID 134873, dilution 1:50) in 1× PBS. This step was followed by a 1 h incubation at RT with an Alexa Fluor 488 donkey anti-rabbit IgG (H+L) antibody (Jackson Immunoresearch, catalog number 711-545-152, dilution 1:200) in 1× PBS, and another incubation for 2 h at RT with a monoclonal mouse anti-human collagen I antibody (Abcam, catalog number ab88147, clone 3G3, dilution 1:100) labeled with an Alexa Fluor 647 antibody labeling kit (Invitrogen, catalog number A20186) following the manufacturer's instructions. Slides were then counterstained with DAPI and mounted using fluorescence mounting medium (Dako, Cat. No. S3023).

**Image acquisition for stained lung tissue samples.** Whole slide scans of HE-stained slides were acquired with a Pannoramic 250 Flash III digital slide scanner (3D Histech, HU). All micrographs from the IF stain were acquired using an LSM980 confocal microscope with Airyscan 2 (Zeiss, DE).

**Generation of TurboGFP-only and TurboGFP-tagged LRRC15 cDNA overexpression lines.** VELOCITY DNA Polymerase was used in PCR to generate amplicons containing cDNA for *TurboGFP*-tagged *LRRC15* Transcript 1 (from Origene plasmid RG225990) and *TurboGFP* control. PCR was similarly used to amplify all components from the LentiCRISPR-v2 plasmid with the exception of the *U6-sgRNA* sequence and *Cas9* protein coding sequence. NEBuilder HiFi DNA Assembly Master Mix was used to assemble lentiviral *LRRC15-TurboGFP* and *TurboGFP*-only cDNA constructs using the cDNA amplicons as inserts and LentiCRISPR-v2 fragment as the vector backbone. Assembly products were transformed into 10-beta cells via heat shock, plated onto agarose containing ampicillin at a concentration of 100 μg/mL and incubated for approximately 16 h at 37°C. Individual colonies were picked, expanded in Luria broth (Life Technologies) supplemented with ampicillin and amplified constructs were harvested using ISOLATE II Plasmid Mini Kit (Bioline). Successful construct assembly was confirmed via Sanger Sequencing. Lentivirus production and transduction of IMR90 fibroblasts and HEK293T-*ACE2-TMPRSS2* cells was carried out as previously described. Cells were then selected on puromycin at a concentration of 2 μg/mL for a minimum of 72 h and functional cDNA expression confirmed by observation of fluorescence.

**SARS-CoV-2 pseudotyped lentivirus production and infection assay.** SARS-CoV-2 pseudovirus was produced using a five-component plasmid system. Plasmid encoding the SARS-CoV-2 spike protein with an 18 amino acid truncation of the C-terminus or the Delta variant of the SARS-CoV-2 spike protein was co-transfected into HEK293T cells with pBCKS (HIV-1SDmCMBeGFP-P2A-luc2pre-IIU), which permits equimolar expression of firefly luciferase and EGFP, and packaging plasmids pHCMVgagpolmllstwhv, pcDNA3.1tat101ml and pHCMVRevmlwhvpre. Transfection was carried out using Lipofectamine 3000 Transfection Reagent (Thermo Scientific) according to manufacturer's instructions. Approximately 16 h after transfection, a media change was performed. Viral media was collected the following day, passed through a 0.45 μm filter and then concentrated using 100K MWCO Pierce Protein Concentrators (Life Technologies Australia). Concentrated virus was then stored at −80°C. Pseudovirus particle concentrations were determined using the QuickTiter Lentivirus Titer Kit (Cell Biolabs) under manufacturer's conditions.

For infection of cells with SARS-CoV-2 pseudovirus in *cis* inhibition assays, WT HEK293T, HEK293T-*ACE2*, and HEK293T-*ACE2-TMPRSS2* cells were transfected with cDNA for *myc-DDK*-tagged *LRRC15* transcript 1 or a control plasmid (lentiGuide-Puro; Addgene #52963). Cells were seeded in 96-well plates, concentrated pseudovirus was added 24 h later in the presence of 8 μg/ml Polybrene. Successful transduction of cells was confirmed by observing GFP expression 48 h post-tranduction. The extent of transduction was quantified with the Steady-Glo Luciferase Assay System (Promega Corporation) according to the manufacturer's instructions. Briefly, plates were allowed to equilibrate to room temperature before 50 μL of Steady-Glo reagent was added to each well containing 50 μL of cell culture media. Plates were incubated at room temperature for 1 h to permit cell lysis and luminescence was then measured using a plate reader. Luminescence of the *LRRC15* cDNA- and control plasmid-transfected cells was normalized to the luminescence values for control cells infected at corresponding viral concentration/pseudovirus number.

**SARS-CoV-2 authentic virus infection assays.**   For assessing the inhibitory effect of native overexpression of LRRC15, HEK293T-*ACE2-TMPRSS2* cells were transfected with *myc-DDK*-tagged *LRRC15* transcript 1 plasmid (Origene, RC225990) for transient overexpression, with empty *myc-DDK* plasmid as a control plasmid. HEK293T-*ACE2-TMPRSS2* cells were seeded in 384-well plates at a density of $8 \times 10^3$ cells/well in the presence of NucBlue live nuclear dye (Invitrogen, United States of America) at a final concentration of 2.5% v/v. The SARS-CoV-2 isolates (Wuhan) were serially diluted in cell-culture medium and an equal volume was then added to the pre-plated and nuclear-stained cells to obtain the desired MOI doses. Viral dilutions were performed in duplicate. Plates were then incubated at 37˚C for 48 h before whole wells were imaged with an IN Cell Analyzer HS2500 high-content microscopy system (Cytiva). Nuclei counts were obtained with automated IN Carta Image Analysis Software (Cytiva) to determine the percentage of surviving cells compared to uninfected controls. *LRRC15* and control plasmid-transfected cells were normalized to the average cell count of uninfected wells for the corresponding cell type to determine the extent of normalized cell death.

**Single-cell RNA-sequencing analysis.**   *LRRC15* expression was first queried on the COVID-19 cell atlas interactive website and summarized in **Fig 5A**. In depth analysis of lung single-cell datasets were conducted on 3 studies [24–26] with Seurat V4.1.0 [60]. Two single-nucleus RNA seq datasets were downloaded from the Single Cell Portal (Broad Institute, SCP1052 and SCP1219) and 1 single-cell RNA seq dataset from Gene Expression Omnibus (GSE158127). Their accompanying metadata, which includes information such as sample ID, sample status, and cluster annotations (cell types), were added to Seurat objects using the "AddMetaData" function. Read counts were normalized using SCTransform, before reanalysis with the standard Seurat workflow of "RunPCA," "FindNeighbours," "FindClusters," and "RunUMAP." Cluster identities were assigned using published cluster annotations and plots were generated with "DimPlot" and "DotPlot." The number of cells in each cluster from each study was then tabulated. "Subset" was utilized to create new fibroblast only datasets before generating collagen (*COL1A1*, *COL1A2*, *COL8A1*, *COL11A1*, *COL12A1*) dotplots for *LRRC15*-expressing (*LRRC15*>0, Pos) and non-expressing (*LRRC15* = 0, Neg) fibroblasts.

**Fibroblast infectivity and SARS-CoV-2 pseudovirus co-culture assay.**   *LRRC15* expression in IMR90 lung fibroblasts were first compared with HEK293T cells by RT-qPCR. These cells were then transfected with empty *TurboGFP* control and *LRRC15-TurboGFP* (Lipofectamine LTX with plus reagent (Thermo Scientific)). Cells were checked for spike-binding activity by incubation with Spike647 and detection via flow cytometry 24 h post-transduction. Then, these fibroblasts were infected with SARS-CoV-2 pseudovirus as described above and luciferase luminescence were compared to HEK293T-*ACE2-TMPRSS2* cells.

For SARS-CoV-2 pseudovirus co-culture assay, IMR90 fibroblasts stably over-expressing either *TurboGFP*-alone or *TurboGFP*-tagged *LRRC15* Transcript 1 were mixed with HEK293T-*ACE2-TMPRSS2* in a ratio of 2:1 and then seeded at a density of 18,000 cells per well in 96-well plates. SARS-CoV-2 pseudovirus was added to cells the following day in fresh media containing 8 ug/mL Polybrene. Extent of transduction was quantified approximately 16 h later using the Steady-Glo Luciferase Assay System (Promega Corporation) as previously described.

**Authentic SARS-CoV-2 virus bioassay of co-cultured cells.** For assessing virus infectivity in the presence of native LRRC15 over-expression, co-culture conditions were established by mixing IMR90 fibroblasts that stably over-expressed *TurboGFP*-tagged *LRRC15* transcript 1 or *TurboGFP*-only with HEK293T-*ACE2-TMPRSS2* at a ratio of 2:1. The cell suspensions were seeded on 96-well plates at a density of 18,000 cells per well. SARS-CoV-2 isolates (Wuhan) were serially diluted in culture medium and an equal volume was added to seeded cells. Plates were incubated for 24 h at 37˚C, and the media was collected and diluted 1:10. This media was added using equal volumes in an infection bioassay consisting of hyperpermissive HEK293T-*ACE2-TMPRSS2* cells that were seeded in 384-well plates (3,000 cells/well). Plates were incubated for 72 h at 37˚C, and NucBlue live nuclear dye (Invitrogen, USA) at a final concentration of 2.5% was added. After a 4 h incubation, plates were imaged using an IN Cell Analyzer HS2500 high-content microscopy system (Cytiva). Quantification of nuclei was performed with automated IN Carta Image Analysis Software and normalized to uninfected wells.

**RNA sequencing.** Total RNA was extracted from IMR90 fibroblasts overexpressing *TurboGFP* alone or *TurboGFP*-tagged *LRRC15* Transcript 1 using the ISOLATE II RNA Mini Kit (Bioline) and quantified via Qubit. A total of 200 ng of each sample was processed with the Illumina Stranded mRNA Prep kit and indexes added with the IDT for Illumina RNA UD Indexes Set A. The prepared libraries were quantified via Qubit and then pooled at a final concentration of 750 pM. PhiX was spiked in at 2% and the pooled libraries were then sequenced on the Illumina NextSeq 2000.

**Differential gene expression analysis and Ingenuity Pathway Analysis.** Differential gene expression analysis was performed using Illumina BaseSpace. Briefly, the BCL Convert app (v2.1.0) was used to generate fastq files from the sequencing run. The DRAGEN FASTQ Toolkit app (v1.0.0) was used to trim adapter sequences and the 5′ T-overhang generated during adapter ligation. DRAGEN FastQC + MultiQC (v3.9.5) was used for quality control checks. DRAGEN RNA (v3.10.4) was used for read counting with hg38 Alt-Masked v2, Graph Enabled used as the reference genome. Finally, DRAGEN Differential Expression (v3.10.5) was used for differential gene expression analysis. Output can be found in **S5 Table**. Output of this differential gene expression analysis was uploaded to Ingenuity (v01-21-03) and filtered such that only genes with adjusted *p*-value < 0.05 were used in core analysis. Canonical pathway analysis output can be found in **S6 Table**. Canonical pathways that had a Z-score of 0 or no activity pattern available were disregarded.

**Quantification and statistical analysis.** SARS-CoV-2 spike glycoprotein titration experiments were analyzed on GraphPad Prism and fitted with nonlinear regression (1 site—specific binding) to identify maximal binding ($B_{max}$) and dissociation constants ($K_D$). CRISPR activation screen analysis was performed using MAGeCK (v0.5.9.2) [7]. For each sample, Z-scores were calculated using normalized read counts. Volcano plots were generated using the EnhancedVolcano package for R. All other plots were generated using ggplot2 or GraphPad Prism. All flow cytometry data was analyzed using FlowJo. All RT-qPCR results were analyzed using $\Delta\Delta C_T$ method. For *LRRC15* and *COL1A1* RT-qPCR in TGFβ-treated cells, *LRRC15* and *COL1A1* expression in each sample was normalized to expression in untreated cells.

Significance was assessed with Mann–Whitney one tailed *t* test. For SARS-CoV-2 pseudovirus and live virus experiments, data shown reflects ≥3 independent replicates. For pseudovirus experiments in monocultures, results are reported as either raw luminescence values or as normalized level of transduction, calculated by dividing luminescence recorded for LRRC15-transfected cells by luminescence relative to control. For pseudovirus co-culture experiments, normalized level of transduction was calculated by dividing luminescence recorded for LRRC15 expressing cells by control cells transduced at the same number of pseudovirus particles. For authentic virus monoculture and co-culture infection assays, cell death for both control and LRRC15-transfected cells was normalized to uninfected cells of the same line. Significance for SARS-CoV-2 pseudotyped lentivirus and authentic virus experiments were analyzed with two-way ANOVA with Sidak multiple comparisons test. For analysis of pooled independent single-cell/nuc sequencing datasets, significance was assessed using unpaired *t* test. Significance of differentially expressed genes (DEGs) was assessed using the DRAGEN Differential Expression application, which utilizes DESeq2. Z-score for DEG canonical pathways was determined by Ingenuity. For RT-qPCR confirmation of up-regulated antiviral gene signature and down-regulated collagen gene signature, expression was normalized to the average of the control GFP cells and significance was assessed using one-tailed Mann–Whitney tests. All error bars in this manuscript report SEM unless otherwise stated.

## Supporting information

**S1 Fig. CRISPR activation screen setup. (A)** RT-qPCR of *ACE2* expression in SAM clonal cell lines transduced with *ACE2* sgRNAs or with HEK293T-*ACE2* cells. Results calculated using $\Delta\Delta C_T$ method and normalized to NTC sgRNA-transduced HEK293T-CRISPRa cells. **(B)** FACS gating strategy. Cells were first gated by forward (FSC) and side scatter (SSC) before filtering for singlets. Spike488 fluorescence was gated by comparison with NTC sgRNA transduced cells. Similar strategy was applied to all flow cytometry experiments. **(C)** FACS results for 3 whole-genome CRISPRa screens with NTC sgRNA-transduced cells as negative controls. For screen 1, cells were incubated with Alexa Fluor 488-conjugated SARS-CoV-2 HexaPro spike (Addgene #154754) and selected on puromycin for 3 days. For screen 2, cells were incubated with Alexa Fluor 488-conjugated SARS-CoV-2 Spike glycoprotein (residues 1–1208, complete ectodomain; gift from Dr. Florian Krammer) and selected on puromycin for 3 days. For screen 3, cells were incubated with Alexa Fluor 488-conjugated SARS-CoV-2 HexaPro spike (Addgene #15474) and selected on puromycin for 8 days. The data underlying all panels in this figure can be found in DOI: 10.5281/zenodo.7416876. (TIFF)

**S2 Fig. CRISPR screen analysis and validation. (A and B)** Gene enrichment analysis of screens 2 **(A)** and 3 **(B)** performed using MAGeCK. Horizontal dotted line indicates *p*-value = 0.05. Vertical dotted line indicates $\log_2$ fold changes of 2. *P*-values and LFCs for all genes in screens 2 and 3 are reported in **S1 Table. (C-D)** Density plot of Z-score (gray) for all sgRNA in **(C)** screen 2 and **(D)** screen 3. Blue vertical lines indicate Z-score for *ACE2* sgRNAs. Red vertical lines indicate Z-score for *LRRC15* sgRNAs. Z-scores calculated as described in methods. **(E)** $\log_2$ fold changes of all genes in screen 1 vs. $\log_2$ fold changes of all genes in screen 2. **(F)** $\log_2$ fold changes of all genes in screen 1 vs. $\log_2$ fold changes of all genes in screen 3. **(G)** *LRRC15* expression of cells in **Fig 2E** quantified via RT-qPCR. **(H)** *ACE2* expression was not increased in *LRRC15* sgRNA transduced cells (quantified via RT-qPCR). **(I)** The 3 sgRNAs for *ACE2* from the Calabrese library used in our screens were transduced into HEK293T-CRISPRa cells, and *ACE2* expression was confirmed via qPCR. Only sgRNA3 induced upregulation in *ACE2* expression. **(J)** Transduced cells in **(I)** were incubated with

Spike647 and analyzed via flow cytometry. Only *ACE2* sgRNA3 cells showed a significant increase in Spike647 binding. The data underlying all panels in this figure can be found in DOI: 10.5281/zenodo.7416876.
(TIFF)

**S3 Fig. LRRC15 is related to TLRs and interacts with spike. (A)** Full phylogenetic tree of LRR-Tollkin family of proteins (includes fly and worm orthologs). **(B)** Co-immunoprecipitation of spike was observed in LRRC15-GFP (transcripts 1 and 2) and ACE2 expressing cells but not in control GFP cells. I = input, FT = flow-through, E = elute. **(C)** Control rabbit IgG did not immunoprecipitate LRRC15 or spike. The data underlying all panels in this figure can be found in DOI: 10.5281/zenodo.7416876.
(TIFF)

**S4 Fig. LRRC15 expression inhibits SARS-CoV-2 spike pseudovirus infection in ACE2 expressing cells. (A)** SARS-CoV-2 pseudovirus carrying a firefly luciferase cassette was applied to HEK293T, HEK293T-*ACE2*, and HEK293T-*ACE2-TMPRSS2* cells for 24 h before luminescence quantification. HEK293T cells were relatively resistant to infection, while HEK293T-*ACE2* and HEK293T-*ACE2-TMPRSS2* expressing cells were infectable. $N = 3$ for each condition. **(B)** Pseudovirus added to ACE2-expressing cells in the context of LRRC15. Titration of $15 \times 10^6$, $62.5 \times 10^6$, $250 \times 10^6$, and $1,000 \times 10^6$ lentiviral particles in HEK293T-*ACE2* cells transfected with 0, 156.25, 312.5, 625, 1,250, and 2,500 ng of *Myc-DDK-tagged LRRC15* plasmid DNA. $N = 2$ for each condition. **(C and D)** Luciferase assay for quantification of SARS-CoV-2 pseudovirus infection in **(C)** HEK293T-*ACE2* and **(D)** HEK293T-*ACE2-TMPRSS2* ($N = 3$). Cells were transfected with plasmid encoding *Myc-DDK-tagged LRRC15* transcript 1 or empty vector as a control. Luminescence for *LRRC15* cells were normalized to control cells. Significance was determined by two-way ANOVA, Sidak multiple comparison test; $****p < 0.0001$, $***p < 0.001$, $**p < 0.01$, $*p < 0.05$. **(E)** Quantification of cell survival after incubation with authentic SARS-CoV-2 virus in HEK293T-*ACE2-TMPRSS2* cells transfected with plasmid encoding *Myc-DDK*-tagged *LRRC15* transcript 1 or *Myc-DDK* only ($N = 3$). Significance was determined by two-way ANOVA, $*p < 0.05$. The data underlying all panels in this figure can be found in DOI: 10.5281/zenodo.7416876.
(TIFF)

**S5 Fig. Single-cell/nucleus analysis of different studies corroborates restricted *LRRC15* expression in fibroblasts. (A)** UMAP plot of lung single-nucleus RNA seq dataset (Delorey and colleagues). **(B)** Feature plot and **(C)** dotplot shows *LRRC15* is expressed in Delorey and colleagues fibroblasts. **(D)** UMAP plot of lung single-nucleus RNA seq dataset (Bharat and colleagues). **(E)** Feature plot and **(F)** dotplot shows *LRRC15* is expressed in Bharat and colleagues lymphatic endothelial cells and various populations of fibroblasts. The data underlying all panels in this figure can be found in DOI: 10.5281/zenodo.7416876.
(TIFF)

**S6 Fig. H&E and immunofluorescence staining of human lung tissue samples.** Representative micrographs of **(A)** hematoxylin and eosin staining and **(B)** immunofluorescence staining of post-mortem formalin-fixed paraffin embedded human lung samples obtained from donors who died of severe COVID-19. Control samples were obtained from patients with melanoma metastases in the lung and non-tumor tissue used for comparisons. Micrographs taken at 200× magnification. Scale bar = 50 μm for **(A)**, 100 μm for **(B)**. In **(B)**, red = Collagen I, green = LRRC15, blue = DAPI. The data underlying all panels in this figure can be found in DOI: 10.5281/zenodo.7416876.
(TIFF)

**S7 Fig. IMR90 fibroblasts bind SARS-CoV-2 spike and *LRRC15* over-expressing fibroblasts show decreased Collagen VI expression. (A)** Representative flow cytometry analysis of IMR90 fibroblasts incubated with Spike647 show cells have intrinsic spike-binding activity (*N* = 2). **(B)** Full images for LRRC15 and Collagen VI western blots. *LRRC15* overexpression in fibroblasts results in decreased Collagen VI protein expression. The data underlying all panels in this figure can be found in DOI: 10.5281/zenodo.7416876.
(TIFF)

**S1 Table. CRISPR activation screen MAGeCK outputs.** Collated output of MAGeCK and MAGeCKFlute pipeline. For each screen, normalized read counts and Z-scores, gene-level summary, sgRNA-level summary, and output of MAGeCKFlute ReadRRA() function are provided.
(XLSX)

**S2 Table. Oligonucleotides for CRISPR activation sgRNA constructs.** Lists oligonucleotides used for generation of CRISPRa sgRNA constructs. Sequences for each sgRNA construct were from either Weismann lab Human Genome-wide CRISPRa-v2 Library (Addgene, #83978) or Calabrese Library Set A (Addgene, #92379).
(DOCX)

**S3 Table. Next-generation sequencing primers.** List of primers used for next-generation sequencing of gDNA extracted from pooled CRISPR activation screen samples. Primers were adapted from Sanson and colleagues [5].
(DOCX)

**S4 Table. RT-qPCR primer sequences.** List of primers used for RT-qPCR.
(DOCX)

**S5 Table. Differential gene expression analysis results.** Output of DRAGEN Differential Expression application.
(CSV)

**S6 Table. Ingenuity comparison analysis canonical pathways results.** Canonical pathways output from Ingenuity Comparison Analysis using DRAGEN Differential Expression results for IMR90 *TurboGFP* vs. IMR90 *LRRC15 T1-TurboGFP* for input.
(XLS)

**S1 Raw images. Raw blot images for Fig 3I, S3B and S3C and Fig 6F.**
(PDF)

## Acknowledgments

We thank Novogene for CRISPRa library sequencing, Sydney Informatics Hub (Artemis HPC) for single-cell data analysis infrastructure, Sydney Cytometry for flow cytometry and FACS support, the technical and scientific assistance of Sydney Microscopy & Microanalysis, the University of Sydney node of Microscopy Australia, Dr. Megan Steain, Dr. Mark Larance, Dr. Sean Humphrey, Dr. Gang Liu, Dr. Phil Hansboro, Dr. Tim Newsome, and members of the Neely lab for helpful discussions. Figure illustrations were created with BioRender.com.

## Author Contributions

**Conceptualization:** Lipin Loo, Matthew A. Waller, G. Gregory Neely.

**Data curation:** Lipin Loo, Matthew A. Waller, Alexander J. Cole, Alberto Ospina Stella, Omar Hasan Ali.

**Formal analysis:** Lipin Loo, Matthew A. Waller, Alexander J. Cole, Alberto Ospina Stella, Omar Hasan Ali, Zina Hamoudi, Wolfram Jochum, Daniel Hesselson.

**Funding acquisition:** G. Gregory Neely.

**Investigation:** Lipin Loo, Matthew A. Waller, Cesar L. Moreno, Alexander J. Cole, Alberto Ospina Stella, Oltin-Tiberiu Pop, Ann-Kristin Jochum, Omar Hasan Ali, Christopher E. Denes, Zina Hamoudi, Felicity Chung, Anupriya Aggarwal, Wolfram Jochum, Lukas Flatz, Daniel Hesselson, Stuart Turville, G. Gregory Neely.

**Methodology:** Lipin Loo, Matthew A. Waller, Cesar L. Moreno, Alexander J. Cole, Alberto Ospina Stella, Oltin-Tiberiu Pop, Ann-Kristin Jochum, Christopher E. Denes, Zina Hamoudi, Felicity Chung, Anupriya Aggarwal, Jason K. K. Low, Karishma Patel, Rezwan Siddiquee, Taeyoung Kang, Suresh Mathivanan, Joel P. Mackay, Lukas Flatz, Daniel Hesselson, Stuart Turville, G. Gregory Neely.

**Project administration:** Daniel Hesselson, Stuart Turville, G. Gregory Neely.

**Resources:** Wolfram Jochum, Lukas Flatz, Stuart Turville, G. Gregory Neely.

**Supervision:** Lipin Loo, Wolfram Jochum, Daniel Hesselson, Stuart Turville, G. Gregory Neely.

**Validation:** Lipin Loo, Matthew A. Waller, Cesar L. Moreno, Alexander J. Cole, Alberto Ospina Stella, Oltin-Tiberiu Pop, Ann-Kristin Jochum, Christopher E. Denes, Zina Hamoudi, Wolfram Jochum, Lukas Flatz.

**Visualization:** Oltin-Tiberiu Pop, Ann-Kristin Jochum, Wolfram Jochum, Lukas Flatz.

**Writing – original draft:** Lipin Loo, Matthew A. Waller, Daniel Hesselson, G. Gregory Neely.

**Writing – review & editing:** Lipin Loo, Matthew A. Waller, Cesar L. Moreno, G. Gregory Neely.

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
