## [Editor Report · Decision Letter 0]

29 Nov 2022

Dear Dr. Neely, 

Thank you for submitting your manuscript entitled "Fibroblast-expressed LRRC15 suppresses SARS-CoV-2 infection and controls antiviral and antifibrotic transcriptional programs." for consideration as a Research Article by PLOS Biology.

Your manuscript has now been evaluated by the PLOS Biology editorial staff, as well as by academic editors with relevant expertise, and I am writing to let you know that we would like to proceed towards publication of your manuscript without further peer review.

However, before we can proceed to the next step, we need you to complete your submission by providing the metadata for your manuscript. To this end, please login to Editorial Manager where you will find the paper in the 'Submissions Needing Revisions' folder on your homepage. Please click 'Revise Submission' from the Action Links and complete all additional questions in the submission questionnaire.

Once your full submission is complete, your paper will undergo a series of checks and after your manuscript has passed the checks, we will send you a decision letter with detailed information. To provide the metadata for your submission, please Login to Editorial Manager (https://www.editorialmanager.com/pbiology) within two working days, i.e. by Dec 01 2022 11:59PM.

Kind regards,

Paula

---

Senior Editor

PLOS Biology

---

## [Editor Report · Decision Letter 1]

6 Dec 2022

Dear Dr. Neely,

Thank you for your patience while your manuscript "Fibroblast-expressed LRRC15 is a receptor for SARS-CoV-2 spike and controls antiviral and antifibrotic transcriptional programs." was being assessedat PLOS Biology. It has now been evaluated by the PLOS Biology editors and by an Academic Editor with relevant expertise.

Based on our Academic Editor's assessment of your revision from the previously reviewed manuscript, we are likely to accept this manuscript for publication, provided you satisfactorily address the following data and other policy-related requests.

**1.** DATA POLICY:

A) Supplementary files (e.g., excel). Please ensure that all data files are uploaded as 'Supporting Information' and are invariably referred to (in the manuscript, figure legends, and the Description field when uploading your files) using the following format verbatim: S1 Data, S2 Data, etc. Multiple panels of a single or even several figures can be included as multiple sheets in one excel file that is saved using exactly the following convention: S1_Data.xlsx (using an underscore).

B) Deposition in a publicly available repository. Please also provide the accession code or a reviewer link so that we may view your data before publication.

Regardless of the method selected, please ensure that you provide the individual numerical values that underlie the summary data displayed in the following figure panels as they are essential for readers to assess your analysis and to reproduce it: Figures 2BCDF, 3C, 4BCDG, 5ABCEFGH, 6ABCDE, and Supplementary Figures SF1A, SF2ABCDEFGHI, SF3A, SF4ABCDE, SF5ABCDEF.

**Please also ensure that figure legends in your manuscript include information on where the underlying data can be found, and ensure your supplemental data file/s has a legend.**

**2.** BLOT AND GEL REPORTING REQUIREMENTS:

We require the original, uncropped and minimally adjusted images supporting all blot and gel results reported in an article's figures or Supporting Information files. We will require these files before a manuscript can be accepted so please prepare and upload them now. We expect this for figures 3I, 6F, and Supplementary Figures SF3BC, SF7B.

Please carefully read our guidelines for how to prepare and upload this data: https://journals.plos.org/plosbiology/s/figures#loc-blot-and-gel-reporting-requirements

**3.** Please add size bars to the microscopy pictures in Supplementary figure SF6AB.

We expect to receive your revised manuscript within two weeks.

*Published Peer Review History*

*Press*

Sincerely,

Paula

---

Senior Editor,

pjaureguionieva@plos.org,

PLOS Biology

---

## [Editor Report · Decision Letter 2]

16 Dec 2022

Dear Dr Neely,

Thank you for the submission of your revised Research Article "Fibroblast-expressed LRRC15 is a receptor for SARS-CoV-2 spike and controls antiviral and antifibrotic transcriptional programs." for publication in PLOS Biology. On behalf of my colleagues and the Academic Editor, Ken Cadwell, I am pleased to say that we can in principle accept your manuscript for publication, provided you address any remaining formatting and reporting issues. These will be detailed in an email you should receive within 2-3 business days from our colleagues in the journal operations team; no action is required from you until then. Please note that we will not be able to formally accept your manuscript and schedule it for publication until you have completed any requested changes.

PRESS

Sincerely, 

Paula

---

Senior Editor

PLOS Biology
